# Coherent two-dimensional electronic mass spectrometry

Sebastian Roeding[1] & Tobias Brixner [1]

Coherent two-dimensional (2D) optical spectroscopy has revolutionized our ability to probe many types of couplings and ultrafast dynamics in complex quantum systems. The dynamics and function of any quantum system strongly depend on couplings to the environment. Thus, studying coherent interactions for different environments remains a topic of tremendous interest. Here we introduce coherent 2D electronic mass spectrometry that allows 2D measurements on effusive molecular beams and thus on quantum systems with minimum system–bath interaction and employ this to identify the major ionization pathway of 3d Rydberg states in $NO_2$. Furthermore, we present 2D spectra of multiphoton ionization, disclosing distinct differences in the nonlinear response functions leading to the ionization products. We also realize the equivalent of spectrally resolved transient-absorption measurements without the necessity for acquiring weak absorption changes. Using time-of-flight detection introduces cations as an observable, enabling the 2D spectroscopic study on isolated systems of photophysical and photochemical reactions.

---

[1] Institut für Physikalische und Theoretische Chemie, Universität Würzburg, Am Hubland, 97074 Würzburg, Germany. Correspondence and requests for materials should be addressed to T.B. (email: brixner@phys-chemie.uni-wuerzburg.de)

Traditionally, molecular-beam spectroscopy is used in a pump–probe implementation with photoelectron or -ion detection for the gas-phase investigation of transition states in photochemistry[1], vibrational wavepackets[2–4], or excited-state dynamics of biologically relevant molecules[5–7]. Furthermore, ultracold samples such as helium nanodroplets[8] facilitate, for example, the investigation of multiple-quantum coherences[9, 10]. Other approaches[11] employ coincidence detection and velocity-map imaging to retrieve information about reaction kinetics and energy levels involved, permitting, for example, chiral recognition in the gas phase[12, 13]. Yet observing kinetics alone does not provide the full coherent picture of the ultrafast system dynamics.

In condensed-phase environments, coherent 2D spectroscopy[14–18] can be regarded as an extension to transient absorption by resolving the pump frequency axis via Fourier transformation over the time delay between two pump pulses. Coherent 2D spectroscopy has been shown, among other applications, to reveal information about energy transport in light-harvesting complexes[19–22], electronic dynamics in semiconductors[23–27], structural dynamics[28–30], or reaction channels in photochemistry[31]. Alternatively to coherence-detected techniques[32], action-based 2D spectroscopy relies on incoherent, that is, population-based, signals. Previous realizations of action-based 2D spectroscopy used fluorescence emitted from atomic vapor[33–35], from molecules prepared in solution[36–38], or from micro-structured samples on surfaces[39], external photocurrents in photoemission electron microscopy[40], and internal photocurrents from solid-state samples[24, 25, 27].

Here we suggest and implement molecular-beam coherent 2D electronic mass spectrometry, combining 2D spectroscopy in effusive molecular beams with cation detection for probing the final-state population. Using time-of-flight mass spectrometry, we obtain a 2D spectrum simultaneously for the parent molecule and each of its fragments. With this implementation of 2D electronic spectroscopy, we exemplarily investigate ionization pathways in nitrogen dioxide ($NO_2$) and acquire ion-selective 2D spectra. Coherent 2D spectroscopy on molecular beams can serve as a complementary tool to condensed-phase techniques resolving otherwise congested 2D spectra of transitions in complex systems due to the narrow linewidths of a gas-phase sample. It further enables the investigation of bound–continuum transitions and corresponding lineshapes by detecting the ionization products.

## Results

**Acquisition schemes.** The basic idea of molecular-beam 2D spectroscopy is not to attempt measuring the coherently emitted nonlinear signal within a four-wave-mixing geometry because, at the very low particle densities of a molecular beam, this signal would be much too weak to be useful. Instead, we take up the concept of action-based detection and measure the photo-generated ions as a function of the time delays and relative phases of an excitation pulse sequence. This approach has the advantage of providing both single-particle detection efficiency and fully coherent spectroscopic information despite the employment of an incoherent observable. Additionally, one obtains 2D spectra not only for the parent molecule but simultaneously for all fragments arising after a photochemical reaction.

Specifically, the measurement scheme (Fig. 1) comprises an ultraviolet (UV) pulse with a center wavelength of 267 nm and a visible pulse with a wavelength of 540 nm that are collinearly combined and focused into the interaction region of a time-of-flight mass spectrometer. The visible pulse can be shaped into a collinear four-pulse sequence utilizing a pulse shaper in the beam path, enabling different experimental configurations: coherent 2D spectroscopy using only the visible four-pulse sequence (Fig. 1a), transient 2D spectroscopy combining the UV pulse and the visible probe sequence (Fig. 1b), and conventional pump–probe spectroscopy using a single UV and a single visible pulse (Fig. 1c). The ion yield of each cation in the resulting mass spectrum (Fig. 1d) serves as a reporter of the final-state population after photoexcitation of the gaseous molecules and is recorded as a function of the respective time delays and, in the case of a four-pulse sequence, of the relative phases between the visible pulses (Fig. 1e). The dependence of the ion yield on the relative phases is

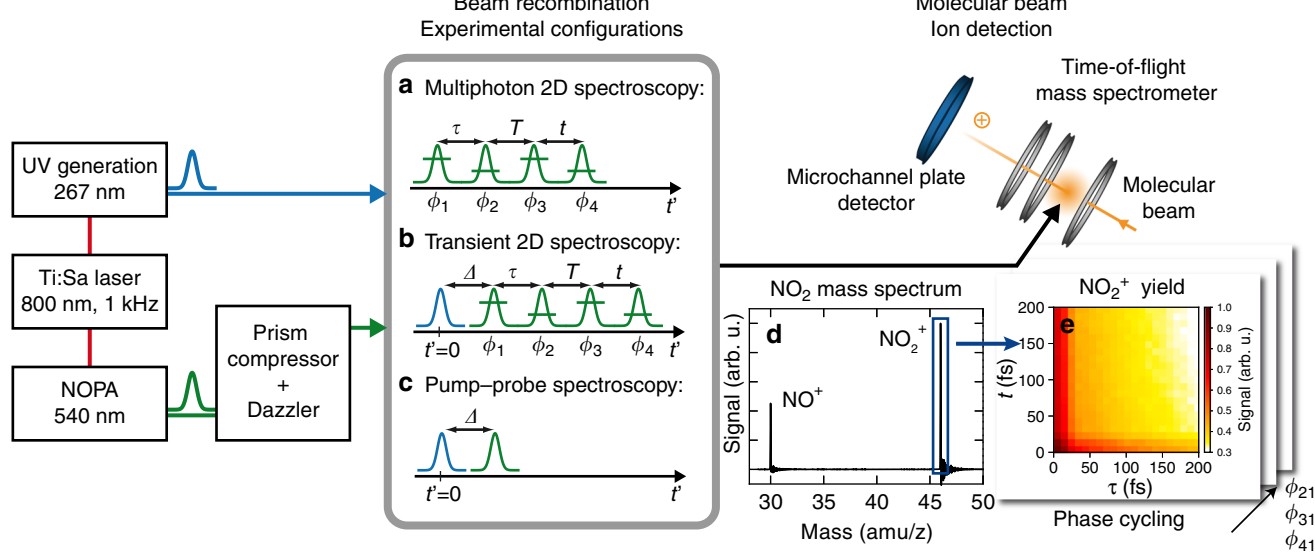

**Fig. 1** Measurement scheme of molecular-beam 2D spectroscopy. Different experimental configurations can be realized, such as **a** multiphoton 2D spectroscopy, **b** transient 2D spectroscopy, and **c** conventional pump–probe spectroscopy. The delay $\Delta$ between both branches can be adjusted by a mechanical delay stage (not shown) in the fundamental beam path of the UV branch. **d** During data acquisition, the individual peaks of the $NO_2$ mass spectrum are integrated (blue box), thereby obtaining **e** the ion yield as a function of coherence times $\tau$ and $t$. The measurement is repeated for different inter-pulse phase combinations (phase cycling) and the nonlinear signal of interest extracted by adding the corresponding maps accordingly (see Methods section)

employed in a 27-step phase-cycling scheme to disentangle different nonlinear signal contributions, such as rephasing (photon-echo) and nonrephasing, to the overall signal during data analysis (see Methods section). 2D spectra are obtained for all types of third-order signal contributions simultaneously from the same raw data set after 2D Fourier transformation, typically with respect to the time delays $\tau$ and $t$.

**Multiphoton-ionization 2D spectroscopy.** We first focus on measurements using only the shaped visible pulse (configuration (a) in Fig. 1), where we simultaneously retrieve the real and imaginary parts of the rephasing 2D spectrum for the parent ion and for the $NO^+$ fragment as well as their absorptive 2D spectra (Fig. 2). The 2D spectra of $NO_2^+$ and $NO^+$ show considerable differences and have amplitudes of opposite signs. Whereas the real part of the $NO_2^+$ rephasing 2D spectrum has a negative peak and displays positive features along the anti-diagonal, the real part of the $NO^+$ signal is positive and elongated along the diagonal with only weak features of opposite sign in the spectrum.

One might speculate whether both the $NO_2^+$ and the $NO^+$ species originate from the same initial excitation pathway, that is, an ionization to $NO_2^+$ which is then followed by dissociation toward $NO^+$ from the same intermediate state. In conventional pump–probe mass spectrometry one could not assess the adequacy of such a hypothesis, as a particular ion-fragment signal might arise from dissociation on a neutral excited-state potential-energy surface followed by ionization (ionization of a dissociation product), or from ionization followed by dissociation in the ionic manifold (dissociative ionization). 2D mass spectrometry offers more insight, even in this multiphoton regime.

If both ions carried the information from the same response function ending in a highly excited final state leading to ionization, one would expect identical signs and the same peak shape in both signals because the $NO_2^+$ and the $NO^+$ yields would just be measures of the same final-state population. In contrast, the observed differences for both ions indicate that the $NO_2^+$ and $NO^+$ 2D spectra likely carry the information from different nonlinear response functions and originate from different

intermediate species. For example, fragmentation can take place at an intermediate level by multiphoton excitation of $NO_2$ within the first two pulses followed by subsequent dissociation during the waiting time $T = 100$ fs and ionization with pulses three and four. This leads to an increased and thus positive $NO^+$ product signal and a depleted, hence negative, $NO_2^+$ reactant signal, similar to product absorption and bleach signals in transient absorption.

According to power-dependent measurements (Supplementary Fig. 1) it requires four visible photons to produce the parent ion and five photons for the fragment. This is in line with previous multiphoton-ionization spectra of $NO_2$ that identified the dominant ionization pathway via a resonantly enhanced step at the three-photon level[41]. Ionization using four photons is accompanied with a change from bent to linear geometry at a resonant intermediate level[42, 43] that can possibly proceed during the waiting time of $T = 100$ fs. This process could in principle be followed by scanning $T$ which was not done here due to a considerable increase in measurement time. Multiphoton ionization can proceed via autoionization[44, 45] that depends on the vibrational quantum numbers and mainly proceeds via the symmetric stretching mode in $NO_2$[46, 47], which was demonstrated in the mode-selective production of $NO_2^+$[48]. Due to the high nonlinearity, the signals in our experiment are dominated by signal contributions of at least eighth order in the number of interactions with the visible laser field that coincide at the same spectral position (for Feynman diagrams of all contributions see Supplementary Fig. 2). Figure 2 demonstrates the simultaneous acquisition of precursor and dissociation-product 2D spectra in a gas-phase photochemical reaction and reveals their individual and distinct nonlinear response functions, which is not feasible using just pump–probe spectroscopy.

**Transient 2D spectroscopy.** Combining the visible four-pulse sequence with a UV excitation pulse (configuration (b) in Fig. 1) allows us to proceed one step further and acquire not only conventional 2D spectra (of effusive molecular beams) but also transient multidimensional spectra[49–51] of the excited $NO_2$

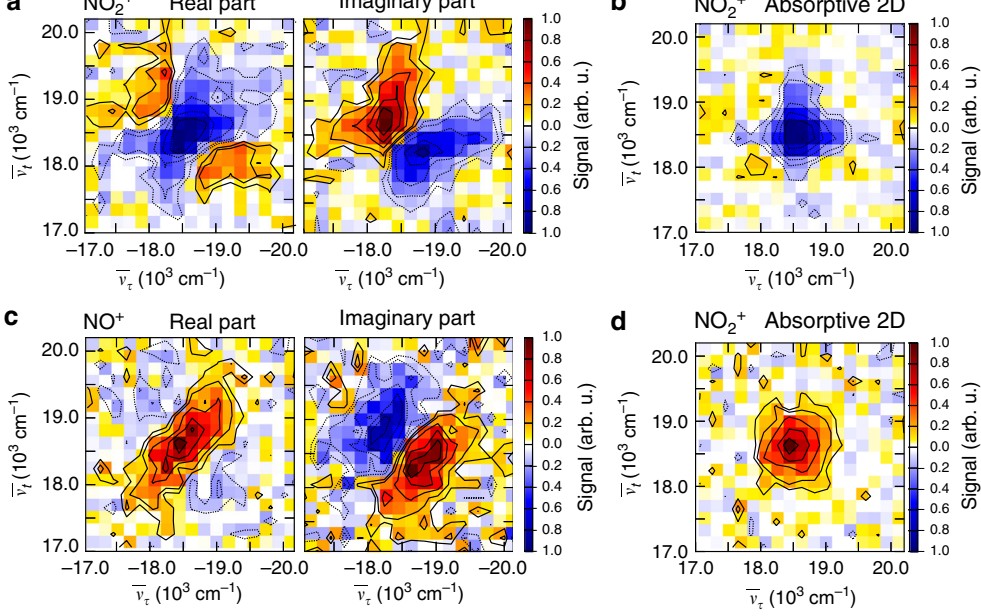

**Fig. 2** 2D spectra using only the visible pulses. **a** Rephasing 2D spectrum for the parent ion with real part (left) and imaginary part (right). **b** Corresponding absorptive 2D spectrum for the parent ion. **c** Rephasing 2D spectrum for the $NO^+$ fragment. **d** Corresponding absorptive 2D spectrum for the $NO^+$ fragment. Nonrephasing 2D spectra and double-sided Feynman are shown in Supplementary Fig. 2

sample for any given delay $\Delta$ (here 150 fs) between the UV pump and the first visible probe pulse. Scanning $\Delta$ retrieves transient information about the evolution of coherences during bond breakage after photoexcitation. As will be detailed in the discussion of pump–probe experiments below, the UV pulse excites the sample by two-photon absorption into 3d Rydberg states that are probed by the interaction with the visible four-pulse sequence, leading to ionization. The UV pulse itself is intense enough to produce a constant level of background ions via (resonant) multiphoton ionization as detailed in Supplementary Note 1. Using phase cycling in the visible probe-pulse sequence and analyzing the data accordingly, transient 2D spectroscopy is, however, only sensitive to signals that arise from the interaction of both, the UV as well as the visible probe pulses and hence automatically discards ion contributions originating only from the UV pulse. Ions generated only by the visible pulses were not detected due to a lower laser intensity in this experiment.

As one particular of the many available contributions, we analyze the rephasing ($\alpha = -1$, $\beta = 1$, $\gamma = 1$, $\delta = -1$) 2D signal (Fig. 3a), with the greek indices $\alpha, \beta, \gamma, \delta \in \mathbb{Z}$ representing the numbers and types of interaction of each of the four probe pulses with the density matrix in double-sided Feynman diagrams[52] (see Methods section). All 2D spectra are shown in a pixelated version, wherein the size of one pixel corresponds to the spectral resolution (159 cm$^{-1}$). The signal originates from four interactions with the pump and four interactions with the probe laser field (Fig. 3b) and is thus relatively weak compared to conventional 2D spectroscopy relying on third-order signals.

Contributions with six or more visible interactions are expected to be even weaker and of no relevance in the case considered here. The real part shows a signal elongated along the diagonal, with the peak shape being characteristic of a sample with large inhomogeneous broadening. For our experimental conditions Doppler and pressure broadening are about four orders of magnitude smaller than the spectral resolution (159 cm$^{-1}$). Thus, we ascribe the elongated signal shape to a multitude of unresolved vibronic transitions between 3d Rydberg states populated by the UV pump pulse and the energetically higher auto-ionizing final states or the ionic continuum reached by the visible probe-pulse sequence. The peaks line up along the diagonal, with a very narrow width across the anti-diagonal as expected from the small linewidth due to the isolated conditions in the gas-phase sample. This leads to an effective peak shape that is much broader in the diagonal than in the anti-diagonal direction.

The two pathways leading to the ionic continuum, direct ionization and absorption into a discrete auto-ionizing state, can interfere giving rise to a Fano-type lineshape of the absorption profile[53]. As demonstrated recently, the lineshape can be controlled and converted back to a Lorentzian absorption profile by an additional laser field[54]. In coherent 2D spectroscopy, Fano resonances lead to distorted peak shapes that depend on the coupling strength between a discrete state and a continuum as analyzed theoretically in a recent work[55] and studied experimentally for molecular layers on plasmonic arrays[56]. We adopted the model of ref.[55] for a qualitative comparison with our data and calculated the lineshape of the rephasing contribution for

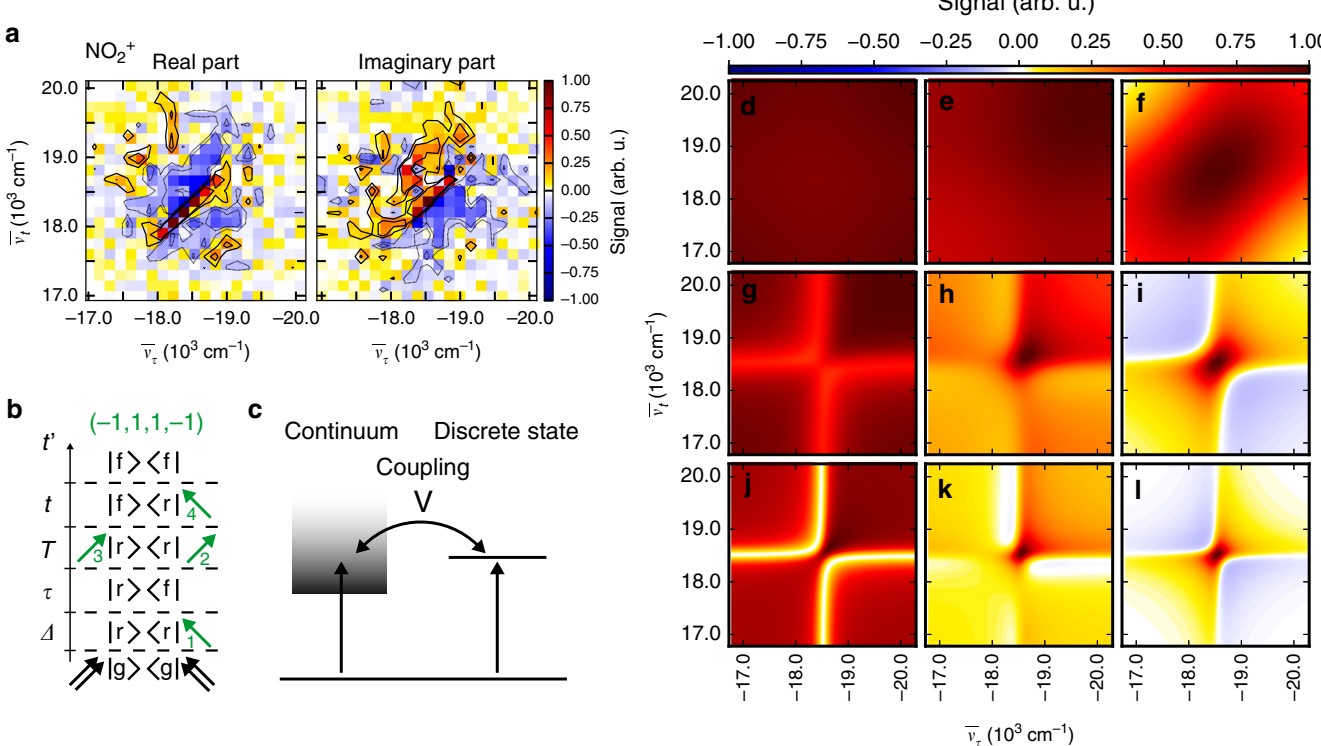

**Fig. 3** Rephasing parent-ion transient 2D spectrum. **a** Experimental real and imaginary 2D spectrum of the 3d Rydberg states prepared by the UV pump pulse. **b** Example of one possible Feynman diagram for the $(-1, 1, 1, -1)$ contribution (a complete list can be found in Supplementary Fig. 3), illustrating the time ordering of interactions (green arrows) for the rephasing signal starting from the ground state $|g\rangle$. The Rydberg states $|r\rangle$ are populated by the UV pump pulse (black arrows); the final state $|f\rangle$ prepared by the probe pulses leads to ionization. **c** Fano model illustrating interference of two pathways to the ionic continuum. The Fano $q$ parameter is inversely proportional to the coupling strength $V$. **d–l** Real part of calculated rephasing 2D spectrum using the 2D Fano model for different values of the Fano $q$ parameter (varied for the individual plots along the vertical assembly direction) and the dissipation parameter $\Gamma$ (along the horizontal direction): $\Gamma = 0.1$, $q = 0.1$ (d), $\Gamma = 0.1$, $q = 1.0$ (e), $\Gamma = 0.1$, $q = 10$ (f), $\Gamma = 0.5$, $q = 0.1$ (g), $\Gamma = 0.5$, $q = 1.0$ (h), $\Gamma = 0.5$, $q = 10$ (i), $\Gamma = 1.0$, $q = 0.1$ (j), $\Gamma = 1.0$, $q = 1.0$ (k), $\Gamma = 1.0$, $q = 10$ (l). The imaginary part is shown in Supplementary Fig. 4

different values of the Fano $q$ parameter, which is inversely proportional to the coupling strength $V$ to the ionic continuum[55] (Fig. 3c). In the case of strong coupling (small $q$) a spectrally broad peak is expected for the rephasing contribution that exhibits an asymmetric lineshape in the direction of the $\bar{\nu}_t = -\bar{\nu}_\tau$ diagonal (Fig. 3d–l). A large $q$ parameter denotes weak coupling and a dominating transition to discrete auto-ionizing states, changing the asymmetric Fano lineshapes into Lorentzian-like lineshapes of discrete states. Additionally, for increasing $\Gamma$ (top to bottom), the dissipation is weaker and the linewidth decreases.

The data shown in Fig. 3a display no asymmetry of this kind and contain a narrow positive signal along the diagonal with a linewidth limited by the experimental frequency resolution that is surrounded by negative features. Comparing this appearance with the 2D spectra in Fig. 3d–l, the agreement is best with a model describing an isolated sample with weak coupling to the ionic continuum ($q = 10$). Thus, we infer a dominating transition from the prepared 3d Rydberg states into a discrete auto-ionizing state. Furthermore, the measured data comprise positive signal contributions at the edges of the spectrum that are not well described within the 2D Fano model in ref.[55]. While this model treats a single auto-ionizing state, we expect multiple auto-ionizing states in the case of $NO_2$. To the best of our knowledge, this case has not yet been elaborated in the literature with respect to lineshapes in coherent 2D spectroscopy and remains an

interesting direction for future studies of bound–continuum transitions in order to get quantitative fits through improved modeling and increased experimental signal-to-noise ratios. The situation is comparable to that in the beginnings of conventional 2D electronic spectroscopy of discrete states 15 years ago. For the first multichromophore measurements, agreements between experiment and theory were mainly qualitative[19]. Even so, it was clear already at the time that 2D electronic spectroscopy is a powerful method offering additional insights compared to conventional transient absorption spectroscopy. With the present implementation of coherent 2D electronic mass spectrometry, we have reached now a comparable first step for the 2D spectroscopy of unbound (ionic continuum), rather than bound (discrete), states and one can see that the method offers a sensitive testing ground for high-level electronic structure and dynamics calculations. Future research on the detailed theoretical treatment of 2D spectra of dissociative processes as well as ionization to continuum states would benefit from, for example, the generalization of wavefunction-based quantum dynamics calculations[57, 58] and a consistent approach to visualize response functions resulting from dissociation processes within Feynman diagrams.

**Transient-absorption spectroscopy.** In the next step, we employ the versatility of the phase-cycling approach and realize

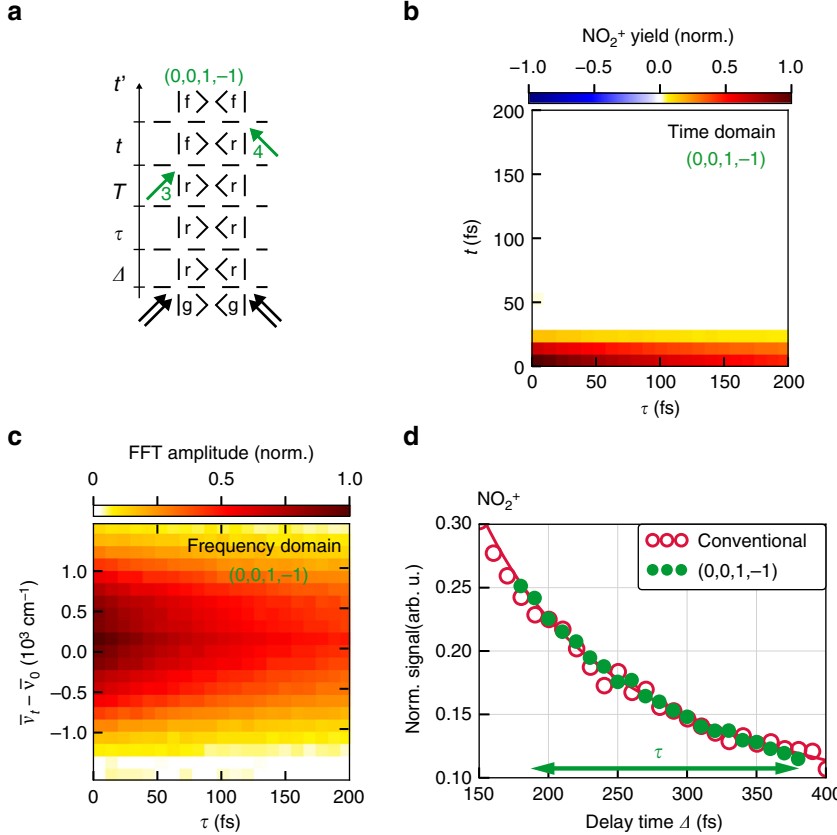

**Fig. 4** Pump–probe contributions obtained from phase cycling. **a** Feynman diagram for the (0, 0, 1, −1) contribution. **b** Time-domain data of the (0, 0, 1, −1) pump–probe contribution. The initial population prepared by the UV pump pulse can decay unperturbedly during coherence time $\tau$. The third probe pulse creates a coherence that evolves as a function of $t$. **c** Fourier transformation over $t$ generates the transient-absorption map of the prepared 3d Rydberg states. **d** Comparison of conventional pump–probe measurement (red circles: data, red line: fit) with pump–probe contribution (0, 0, 1, −1) (green dots) obtained from phase cycling. As the absolute signal levels of the conventional pump–probe trace and the integrated pump–probe contribution retrieved via phase cycling are different, the amplitude of the ($\alpha = 0$, $\beta = 0$, $\gamma = 1$, $\delta = -1$) pump–probe contribution was scaled accordingly. The scaling factor was determined by minimizing the least-square deviation between the ($\alpha = 0$, $\beta = 0$, $\gamma = 1$, $\delta = -1$) signal and the corresponding values of the fit to the conventional pump–probe trace (see below, the fitting was performed beforehand). The effective scanning range for this contribution is indicated by the green arrow

pump–probe-like transient absorption spectroscopy on effusive molecular beams without the need for measuring absorption changes directly. This is highly relevant because absorption changes are minuscule due to the extremely low particle density in molecular beams. Obtaining spectral information from time-dependent measurements in molecular beams has been demonstrated in the 1990s[3, 4, 59, 60]. In those cases, the spectral signature of the transition to the excited state was obtained revealing, for example, vibrational level spacings. In this work, we show how time-dependent absorption can be extracted, that is, in relation to the mentioned pioneering work we add an additional time coordinate and observe how spectral properties evolve. We here extract the desired information from the very same raw data set as the transient 2D rephasing signal. This requires contributions for which two interactions are distributed among the four visible probe pulses. In Fig. 4a we show as an example the Feynman diagram of the ($\alpha = 0$, $\beta = 0$, $\gamma = 1$, $\delta = -1$) contribution for $NO_2^+$, which originates from one interaction with probe pulse 3 ($\gamma = 1$) and one with probe pulse 4 ($\delta = -1$).

Since the first two probe pulses do not interact in this contribution, the initial population prepared by the UV pump pulse (black double arrows, Fig. 4a) decays without additional perturbation by these two pulses during the time intervals $\Delta$, $\tau$, and $T$. The third probe pulse (green arrow, Fig. 4a) generates a coherence $|f\rangle\langle r|$ that evolves during $t$. Finally, the fourth probe pulse in the sequence prepares the final-state population leading to ionization. Accordingly, the extracted signal contribution corresponding to this pump–probe pathway exhibits a population decay along the $\tau$ axis and evolves in a coherence during the time $t$ (Fig. 4b). A one-dimensional Fourier transformation with respect to $t$ can now be applied to retrieve the absorption spectrum at each $\tau$ delay and thus to generate a transient absorption map of the excited 3d Rydberg states (Fig. 4c). Note that obtaining transient absorption data in this fashion is a feature of this work. Using ion detection instead of transient absorption detection, combined with explicit Fourier transformation, provides the desired information. In the exemplary case studied here, the $NO_2^+$ yield comprises a spectrally broad peak and decays as a function of $\tau$, displaying the decay of the population in the two nearly degenerate 5 $^2A_1$ and 6 $^2A_1$ states and thus the dissociation of $NO_2$. The two states cannot be directly resolved, in particular since many transitions from these

two states into the final-state manifold are possible resulting in a congested absorption profile.

For direct comparison of pump–probe data obtained from the visible four-pulse sequence (configuration (b) of Fig. 1) via phase cycling with a conventionally obtained pump–probe data set (without frequency resolution) using just one single pump and one single (ionization) probe pulse (configuration (c) of Fig. 1), the time-domain data (Fig. 4b) is integrated over $t$, providing the green data points in Fig. 4d. For the selected signal contribution, scanning the delay $\tau$ in Fig. 1b corresponds to scanning the delay $\Delta$ in the pump–probe experiment of Fig. 1c, as the molecular system is in the same population state in both configurations. Having set the population time $T = 30$ fs in the four-pulse sequence, the third probe pulse arrives at a total delay of ($\Delta + T$) $= 180$ fs with respect to the UV pump pulse and is then shifted to larger delays by scanning $\tau$. The comparison between both signals shows excellent agreement, confirming the validity of our technique.

The pump–probe data can be recorded and analyzed over a larger range of delays as discussed in detail in Supplementary Note 4. Briefly, we show in Fig. 5(a) the measured pump–probe signal for the $NO_2^+$ parent ion (top, purple dots) and the $NO^+$ fragment (bottom, orange dots) as a function of the delay $\Delta$ between UV and visible pulse. We fit both traces simultaneously using a parallel kinetic rate model[61, 62] resulting for positive delay times (for which the UV acts as a pump) in a decay with time constants of $\tau_1^+ = (60 \pm 10)$ fs, $\tau_2^+ = (264 \pm 47)$ fs (red line), $\tau_3^+ = (3222 \pm 589)$ fs, and a slow decay of several hundred picoseconds for the fragment trace. The observed dynamics are attributed to the excitation into 3d Rydberg states by absorption of two UV photons and subsequent dissociation (Fig. 5b). For negative delay times, a decay with $\tau_1^+ = (141 \pm 106)$ fs and an additional persistent offset is observed for the fragment trace.

## Discussion

In this article we introduced mass-selective molecular-beam multidimensional electronic spectroscopy combining 2D spectroscopy with mass spectrometry. The new action-based variant allows us to investigate the nonlinear response of a low-density gaseous sample. For this we use a phase-cycled sequence of four collinear probe pulses, thereby spreading the information content of transient mass spectrometry over two coherent frequency

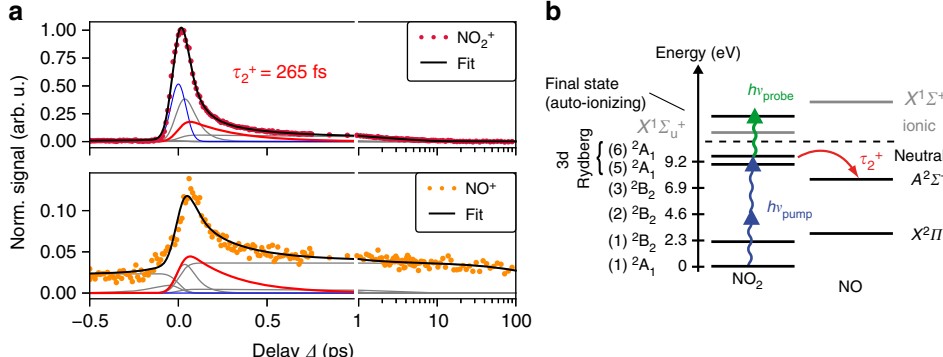

**Fig. 5** Pump–probe mass spectrometry of $NO_2$ using 267 nm pump and 540 nm probe pulses. **a** $NO_2^+$ signal (purple dots, upper diagram) and $NO^+$ signal (orange dots, lower diagram) as a function of pump–probe delay $\Delta$. Positive $\Delta$ signifies that the UV pulse arrives first. Note the logarithmic delay axis for $\Delta$ >1 ps. The black lines indicate the fit of a kinetic rate model to the data, the gray lines show the different contributions to this model (for discussion see Supplementary Note 4). The instrument response function (blue line) is used to account for the ion yield during pulse overlap. The contribution attributed to $NO_2$ dissociation is highlighted (red line). **b** Energy levels of $NO_2$ relevant for the dynamics observed at positive delay times. The UV pulse prepares 3d Rydberg states (corresponding to 5 $^2A_1$ and 6 $^2A_1$) by two-photon absorption (blue arrows) which is probed by the visible pulse (green arrow). This leads to dissociation and production of neutral NO fragments (red arrow). The decay of the parent ion signal with the time constant $\tau_2^+ = 265$ fs is attributed to this dissociation process

dimensions. This generally provides means to dissect transition states and couplings in gas-phase photophysics and photochemistry. Phase cycling enables the retrieval of various nonlinear signal contributions, such as rephasing (photon echo), non-rephasing, or double-quantum coherence.

With coherent 2D electronic mass spectrometry at hand, we investigated the (multiphoton) ionization of $NO_2$, where we identified transitions into auto-ionizing states as the major probe step for 3d Rydberg states pumped by two-photon absorption. Selecting contributions with interactions from one pump pulse and two phase-coherent probe pulses, we obtained frequency-resolved transient-absorption-like maps of a gaseous sample, which cannot be achieved in standard setups measuring the absorption of the transmitted beam due to a lack of sensitivity.

Molecular-beam 2D electronic spectroscopy permits, for example, the experimental investigation of revivals in coherent wave-packet motion[63, 64] and enables to follow the evolution of coherences during bond breakage or isomerization. It further facilitates comparison with complementary liquid-phase 2D spectroscopy based on fluorescence detection[65] by investigating the same molecule in different environments within one experiment, rendering it possible to study in detail the role of the environment during primary steps of photochemistry. This work constitutes a primary example of coherent 2D spectroscopy applied to ionization processes.

## Methods

**Data acquisition**. We use the output of a commercial Ti:Sapphire amplifier system (Spitfire, Spectra Physics) with a pulse duration of 120 fs and a repetition rate of 1 kHz to generate 267 nm (2 nm full width at half maximum, FWHM) ultraviolet (UV) pump pulses via third-harmonic generation. For this, the 800 nm fundamental is frequency-doubled in a 100 μm β-barium borate (BBO) crystal and subsequently mixed with the remaining fundamental beam in an additional 100 μm BBO crystal, delivering a pulse energy of up to 20 μJ. Intensities are controlled using a neutral-density filter wheel in the fundamental beam path. A second branch of the fundamental output is used to pump a commercial non-collinear optical parametric amplifier (TOPAS White, Light Conversion), generating probe pulses centered at 540 nm (34 nm FWHM). This beam passes a single-prism compressor[66] to account for the positive group-delay dispersion introduced by a subsequent acousto-optical programmable dispersive filter (AOPDF) pulse shaper (WR 510, Dazzler, Fastlite) that is capable of performing shot-to-shot phase cycling[33, 67] at the repetition rate of the laser as demonstrated previously with rapid-scan coherent 2D fluorescence spectroscopy[65].

Pump and probe pulses are collinearly recombined using a dichroic mirror and focused into the interaction region of a home-built reflectron-type[68] time-of-flight mass spectrometer (TOF-MS) using an $f = 200$ mm lens. For characterization, a planar mirror is placed in front of the vacuum viewport of the TOF-MS, mapping the focus of both pulses to an accessible region outside the chamber. An identical viewport is inserted in this beam path and characterization is carried out at the external focus position. The visible pulse is compressed to 17 fs by the pulse shaper and characterized via pulse-shaper-assisted collinear frequency-resolved optical gating (cFROG)[69, 70]. The UV pulse has a pulse duration of 130 fs as characterized via xFROG[69] in a difference-frequency-generation process with the compressed visible pulse as the reference. Both pulses are delayed with respect to each other by a computer-controlled delay stage (M-ILS250CCHA, Newport Spectra Physics) in the fundamental beam path of the UV branch.

The $NO_2$ sample is expanded as an effusive beam into the vacuum chamber, increasing the base pressure in the interaction region to $1 \times 10^{-4}$ mbar. In order to avoid contributions of $N_2O_4$ which might be present at room temperature[71], the nozzle is held at a constant temperature of 150 °C by resistive heating. The ion signal is detected on a shot-to-shot basis with a micro-channel-plate detector and recorded with an analog-to-digital converter card (ADQ14, SP Devices Sweden AB) after amplification (Ortec 9302, Gain 20). A typical mass spectrum of the $NO_2$ sample is shown in Fig. 1, displaying the $NO_2^+$ parent ion and the $NO^+$ fragment. Peaks of a smaller mass-to-charge ratio are not observed. For data analysis the mass peaks are integrated over a window of 20 sampling points corresponding to a mass-to-charge ratio of 0.1 amu/z.

In multidimensional experiments using only the visible pulses, the focus intensity is increased via expanding the beam by a factor of two in front of the focusing lens, yielding a maximum intensity of the visible pulse of $2 \times 10^{13}$ W/cm$^2$ when all pulses overlap with equal phase. In this configuration, the visible four-pulse sequence is intense enough to generate ions. The coherence times are scanned from 0 to 150 fs in 10 fs steps at a constant population time of $T = 100$ fs. Averaging is performed 17,230 times. For pump–probe and transient 2D

experiments, the intensities at the sample position are adjusted to $1 \times 10^{13}$ W/cm$^2$ for the (single) visible probe pulse and $3 \times 12^{11}$ W/cm$^2$ for the UV pump pulse.

In transient multidimensional spectroscopy experiments the first probe pulse of the sequence is always kept at a constant delay of $\Delta = 150$ fs with respect to the preceding 267 nm UV pulse. The coherence times $t$ and $\tau$ are scanned from 0 to 200 fs in 10 fs steps, while the population time $T$ is kept constant at $T = 30$ fs. The measurement is repeated 27,000 times for averaging. A comparison with a measurement scanning the identical visible pulse sequence with a photodiode placed in the beam path outside the vacuum chamber as well as the raw experimental 2D spectrum can be found in Supplementary Fig. 3.

In conventional pump–probe experiments the delay $\Delta$ between UV pump and visible probe pulse is scanned from $-5$ to $-500$ fs in 46 steps (100 fs), from $-500$ to 750 fs in 126 steps (10 fs), from 750 to 5 ps in 43 steps (101.2 fs) and additionally from 5 to 100 ps in 41 steps with exponentially increasing step size. Here, $\Delta > 0$ corresponds to the UV pulse acting as pump and the visible pulse as probe pulse. For each time delay the signal is integrated over 2000 laser shots and averaged over 40 complete scans. Since pump and probe beam on their own already generate ions, we use a chopping sequence at each delay position in order to measure the pump-only and the probe-only mass spectrum and thus to extract the pump–probe signal by subtracting both from the mass spectrum when both pulses are present.

In all experiments employing a visible four-pulse sequence, a phase factor $\phi = (1 - \gamma_0)\omega_0 t'$ is added to the probe pulses two, three, and four in order to shift their carrier phases with respect to the envelope depending on the respective delay $t'$ to the first probe pulse. This moves coherent oscillations from optical frequencies (laboratory frame, $\gamma_0 = 1$) to a region close to the origin in frequency space (rotating frame, $\gamma_0 = 0$)[72]. Here we use $\gamma_0 = 0$ and set $\omega_0$ at the center of the probe-pulse spectrum, $\omega_0 = 3.49$ rad fs$^{-1}$ ($\bar{\nu}_0 = 18519$ cm$^{-1}$), to perform the experiments in the rotating frame. Despite this, we plot all 2D spectra with laboratory-frame frequency axes by adding the rotating-frame center frequency to the measured frequencies.

**Data processing**. The total detected fourth-order population $p^{(4)}$ is a sum over the different Liouville pathways[52] $Q^{(4)}(\tau, T, t, \alpha, \beta, \gamma, \delta)$ and depends on the phases $\phi_i$ of the four incident pulses via[67]

$$p^{(4)}\left(\tau, T, t, \phi_1, \phi_2, \phi_3, \phi_4\right) = \sum_{\alpha, \beta, \gamma, \delta} Q^{(4)}(\tau, T, t, \alpha, \beta, \gamma, \delta) e^{i(\alpha\phi_1 + \beta\phi_2 + \gamma\phi_3 + \delta\phi_4)}, \quad (1)$$

with the greek indices $\alpha, \beta, \gamma, \delta \in \mathbb{Z}$ representing the numbers and types of interaction of each of the four probe pulses with the density matrix in double-sided Feynman diagrams[52]. Since the final state is a population state located on the diagonal of the density matrix, the condition $\alpha + \beta + \gamma + \delta = 0$ applies, permitting to consider relative phases $\phi_{i1} = \phi_i - \phi_1$ referenced to the first probe pulse[67]. The contribution of interest, $Q^{(4)}(\tau, T, t, \alpha, \beta, \gamma, \delta)$, can be extracted by acquiring the total signal $p^{(4)}(\tau, T, t, \phi_{21}, \phi_{31}, \phi_{41})$ for different combinations of the relative phases (phase cycling) and summing these measurements,

$$Q^{(4)}(\tau, T, t, \alpha, \beta, \gamma, \delta) = \frac{1}{LMN} \sum_{l=0}^{L-1} \sum_{m=0}^{M-1} \sum_{n=0}^{N-1} p^{(4)}\left(\tau, T, t, l\Delta\phi_{21}, m\Delta\phi_{31}, n\Delta\phi_{41}\right) W_L^{l\beta} W_M^{my} W_N^{n\delta},$$

(2)

with contribution-specific weights $W_L^{l\beta} = e^{-i\beta l \Delta\phi_{21}}$, $W_M^{my} = e^{-iym\Delta\phi_{31}}$, $W_N^{n\delta} = e^{-i\delta n\Delta\phi_{41}}$, and phase increments $\Delta\phi_{21} = \frac{2\pi}{L}$, $\Delta\phi_{31} = \frac{2\pi}{M}$, $\Delta\phi_{41} = \frac{2\pi}{N}$. The $1 \times L \times M \times N = 1 \times 3 \times 3 \times 3$ phase-cycling scheme selected here retrieves, among others, the rephasing ($\alpha = -1, \beta = 1, \gamma = 1, \delta = -1$) and nonrephasing ($\alpha = 1, \beta = -1, \gamma = 1, \delta = -1$) signals for which each visible pulse interacts once with the sample.

**Data availability**. The data that support the findings of this study are available from the corresponding author upon reasonable request.

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

## Acknowledgements
This work was funded by the European Research Council (ERC) within the Consolidator Grant MULTISCOPE. We thank Daniel Finkelstein-Shapiro for fruitful discussions of the 2D Fano model.

## Author contributions
T.B. conceived the experiment. S.R. and T.B. implemented the experiment. S.R. performed the measurements and analyzed the data. Both authors discussed the results and contributed to the manuscript.

## Additional information

**Competing interests:** The authors declare no competing interests.

