## [Peer Review File · Nature Communications]

Reviewers' comments:

Reviewer #1 (Remarks to the Author):

The manuscript title "Molecular-beam coherent 2D electronic spectroscopy reveals photoionization pathways" ("molecular beam" should be unhyphenated) indicates a very promising work. Having 2D coherent spectroscopy available in molecular beam experiments would be a seminal step for a broad spectrum of research. Condensed phase multidimensional spectroscopy is a powerful technique but it is suffering because of unambiguity in the interpretation coming from the often very complex systems which have been studied. This has already led, in particular in Nature and Science papers, to later on revised interpretations e.g. regarding the excitement on quantum effects in biology. Options of having 2D electronic spectroscopy on molecular beams in that respect will be able to answer fundamental questions on electronic processes of molecular systems. Roeding and Brixner have managed to develop a technique for doing this and are presenting first results. This is indeed original new work of high importance and deserves publishing in Nature Communications.

Specific remarks to the manuscript:

"molecular beams" (title and text): In the Methods section one finds "sample is expanded as an effusive beam into the vacuum chamber, increasing the base pressure in the interaction region to 1×10^{-4} mbar". If the experiments probes an increased base pressure in the interaction region one would not call this a molecular beam but an experiment in the gas-phase similar to gas cells. This would still be an interesting important step for 2D spectroscopy but different from molecular beam studies and would not justify the title.

Section "Multiphoton ionization 2D spectroscopy": Here, they demonstrate their method and that they can obtain 2D maps specific to a detection of the parent or fragment ion of gas-phase samples. This is new and original; the data are on multiphoton processes which hampers an easy interpretation. Hence, there is no interpretation of the maps for these results in terms of new insight of the system specific to the 2D method.

Section "Transient 2D spectroscopy": Here the system is pumped by an extra UV pulse at a fixed delay time; no transients are recorded. The results are interpreted as Fano couplings to continuum states. Model calculation are performed and compared to the results. This is indeed a very interesting case where information is obtained specifically to the 2D method. Again, multiphoton processes and overlapping contributions have to be interpreted. The number of involved photons is nicely assigned in power-dependent measurements and the phase-cycling method can select specific interaction paths, demonstrating the advantages of the introduced technique. I have some doubts on the validity of the interpretation in comparison to the modeled distributions because of the low signal to noise ratio. Do the data, shown in Fig. S3a and S5a, represent the same data set? The main features in the center seem to be identical but the noise only in the outer regions appears to be less in S5 (?). For me, just based on Fig. S3, I would not dare to do an interpretation in comparison with the theory. The $q=10$ model plot has in total only very weak negative intensity – the experiment is dominated by negative values. The specific streaks in the imaginary part cannot be identified. The resolution in the direction of the diagonal appears to be very low which is not plausible. I would expect that a stronger dissipation (lower Γ value) could explain this. Here no comparison is shown to justify the chosen value and the interpretation of low dissipation. On the other hand, in the direction perpendicular to the diagonal, the spectral resolution appears to be very high (change from +1 to -1 from one pixel to the next and back). From that I conclude the experimental resolution to be high (according to the 200fs scan width).

Section "transient absorption spectroscopy": I would not call the results of this part a "new type of ...

spectroscopy". The technique represents a one-dimensional Fourier transform spectroscopy which goes in many variants 70 years back to the work of Ramsey. Characterization of electronically excited states with ion detection in the gas-phase has already been done in the 90s (e.g. Chem.Phys.Lett. 233 (1995) 491-499).

In a rate equation model the pump-probe data are fitted. Here one should be careful fitting the 2 transients with 5 parameters, and even for the NO⁺ signal not getting good agreement (increasing slope around 0). The authors should make clear that the numbers and error margins are justified.

In general, the manuscript is very well written, and, in particular, related work is introduced and referenced. The certainly complicated matter of multidimensional spectroscopy of higher order processes is presented in a comprehensible and well-structured way. I would suggest reducing the redundancy of figures.

In conclusion, the manuscript presents a milestone in coherent spectroscopy but needs revisions due to the remarks above. In particular, the interpretation in comparison with the Fano model has to be convincing because this is the scientific achievement obtained by the new method.

Reviewer #2 (Remarks to the Author):

In this manuscript, the authors present a new technique that combines 2D electronic spectroscopy with time-of-flight mass detection. This new technique measures the photogenerated ions as a function of time delays and relative phases of the excitation pulse sequence. The ion yield in the resulting mass spectrum serves as a reporter of the final-state population. By directing both UV and visible beams to the sample and selecting different excitation sequences, the authors are able to perform multi-photon ionization 2D spectroscopy, transient 2D spectroscopy, and transient absorption spectroscopy with the same optical setup. The authors further prove the capability of their new technique by investigating the photodynamics of NO₂. With multiphoton 2D spectroscopy, the authors reveal that the parent ion NO₂⁺ and the fragment ion NO⁺ originate from different intermediate species. With transient absorption spectroscopy, the authors identify transitions from 3d Rydberg states of NO₂ to auto-ionizing states.

The idea of combining 2D electronic spectroscopy is novel and brilliant. I realize that the implementation of this technique is difficult, and I am impressed by the data presented in this manuscript. This new technique renders possibility to detailed studies on the role of the environment and coupling between different states in primary steps of photochemistry. This manuscript will inspire research on both nonlinear spectroscopy and gas phase dynamics and is certainly worthy of publication in Nat. Commun. However, there are a few major issues that the authors need to address regarding to details of their experimental procedure and data interpretation. I therefore recommend publishing this manuscript after revision.

Major comment:

1. The energy requirements to generate NO₂⁺ must be well justified and explained. In the power dependence study presented in SI, the authors claim that the two-color pump-probe experiment requires 2 UV photons and 1 photon of the visible pulse to generate NO₂⁺. But they claim that the 2D experiment requires 4 photons from the visible pulse to generate NO₂⁺. The UV beam in this experiment is centered at 267 nm with pulse duration of 130 fs. The visible beam is centered at 540nm with FWHM of 34 nm and pulse duration of 17 fs. My question is: If 4 visible photons have enough energy to populate the system into auto-ionization states, why 2 UV photons cannot?

The authors claim that the vertical ionization potential of NO₂ of the equilibrium geometry of the ground state is 11.23 eV, but the equilibrium geometry of the molecule at excited state surface can change from a bent configuration to a linear configuration, which is identical to ionization energy of 9.586 eV and can be assessed by 4 visible photons. Then how quick is the change of geometry? The 2D spectrum of NO₂ are taken at T = 100 fs.

If this change of geometry happens within 100 fs, will the 2 UV photons do the same thing because the UV beam has pulse duration of 130 fs?

2. In the multiphoton 2D spectroscopy part, the authors state that the ionization of NO₂⁺ requires 4 visible photons and is in line with previous (1+2+1) multiphoton-ionization process. They also claim that it requires 5 visible photons to generate NO⁺. But what is (are) the detailed ionization pathway(s) in this experiment, (2+2+1) or others? To my knowledge, there are multiple ionization pathways to generate NO⁺ (J. Chem. Phys. 75, 2643–2651 (1981); Annu. Rep. Prog. Chem., Sect. C: Phys. Chem. 106, 274–304 (2010)). Are the NO⁺ coming from dissociation and ionization (NO₂ to NO to NO⁺, as they mentioned in the manuscript, Section multiphoton ionization spectroscopy) or ion-pair state and dissociative ionization (NO₂ to ion-pair state to NO⁺+O⁻, as they mentioned in SI, Section 4)?

3. The authors draw an energy diagram that is relevant to the two-color pump probe experiment in Fig. 5b. However, there are no energy diagrams for the multi-photon 2D measurements or transient 2D measurements in the current manuscript. Different excitation sequences and beams will access different electronic states. Although there are multiple high energy states in NO₂, the electronic structure of NO₂ is well studied in previous literatures (Annu. Rep. Prog. Chem., Sect. C: Phys. Chem. 106, 274–304 (2010); J. Chem. Phys. 115, 10394-10403). The authors should provide energy diagrams for the other two measurements, which can help the data interpretation.

4. The Feynman diagrams in Fig S2.c are misleading. For example, in the first diagram on the top row of Fig S2.c, the first two field interactions lead to the transition from |g> to |e1>. But in the second diagram on the top row of Fig S2.c, the first field interaction leads to the transition from <g| to <e1|. The authors should draw their Feynman diagrams according to the electronic structure of NO₂ because the states have to be accessible with their laser pulses. The nomenclature e1, e2, f should be replaced with actual states of NO₂.

5. Similarly, for the Feynman diagrams with 10th-order contributions that give rise to NO⁺ signals, different ionization pathways will give different Feynman diagrams and it is important to draw the dissociation process if it happens during the waiting time of certain pathways.

6. On page 7 of the main manuscript, second to the last line, the authors state that they scan τ to achieve the time delay between the UV and visible beams. But in the Method section (Page 11), they claim that they scan Δ for the pump-probe measurements. Which experimental parameter is actually scanned in the experiment?

7. In the pump-probe measurement, the authors attribute NO⁺ signal at short delay times to ion pair dissociation (NO⁺ and O⁻) after (2 UV+ 1 Vis) excitation. They have also attribute the long-lived NO⁺ signal to neutral NO₂ dissociation from 3d Rydberg states after 2 UV photon excitation and formation of excited NO in the A Σ^+ state followed by subsequent ionization. But is 1 visible photon enough to ionize NO? Or, this process requires multiple photons? From Fig 5b, it seems like 1 visible photon cannot provide enough energy.

Minor comments:

1. The time delay label in Fig 1 beam configuration c needs to be consistent with the Feynman diagram in Fig 4a.
2. The energy axis needs to be added to Fig. 5b.
3. There is a typo in Figure S2 caption. "nonlinearity that are contributing to the rephasing fourth-order signal shown in Figs. 5(a) and (c)" should be " Figs. 2(a) and (c)" instead.
4. The authors provide data processing and phase cycling equations for fourth-order signals, yet the signals that contribute to the 2D spectrum are 8th and 10th order signals. I recommend the authors to add a few sentences in the SI, explaining how these higher order signals survive with their current phase-cycling scheme and what higher step phase cycling scheme can in principle discriminate between these signals.
5. The authors may consider changing the title of the manuscript. In the current manuscript, the photoionization pathways are identified by the two-color pump probe measurement, not multiphoton or transient 2D spectroscopy. Although I understand these are difficult measurements, it is still too far from claiming the 2D spectra reveal ionization pathways.

In summary, the concept of combining 2D electronic spectroscopy with mass detection is creative and insightful. I am impressed that the authors successfully performed multiple measurements on NO₂ and present high quality data on high order nonlinear signals. This is a tour de force experiment. This manuscript is certainly interesting to nonlinear spectroscopy community and photochemistry/photophysics community. This paper should absolutely be published in Nature Communications after suitable revision.

Reviewer #3 (Remarks to the Author):

The authors present a 2D spectroscopy probe of NO₂ in the gas phase. The work is technically proficient and carefully analyzed. However, this seems to be a technical demonstration of 2D spectroscopy in the gas phase and no new information about NO₂ spectroscopy or dynamics was obtained. Much is known about this molecule and the authors demonstrate no advances compared to this extensive body of existing knowledge. The motivation for 2D gas phase spectroscopy has not been convincingly made: there seems to be no new information obtained from this study. This paper should be re-submitted to a specialist physical chemistry journal. Even then, were I to referee it for this case, I would still insist upon a proper review of the well-known spectroscopy and dynamics of this molecule (see the partial list below) and a clear demonstration of some new advance in understanding. Otherwise, it seems to be a purely technical demonstration without any clear evidence of the need for or advantage of this type of spectroscopy for gas phase dynamics.

The authors seem unaware of extensive, detailed work on NO₂ by S.T. Pratt (Argonne):

Mode-dependent vibrational autoionization of NO₂.
Journal of Chemical Physics 119, 10146 (2003)

Mode dependent vibrational autoionization of Rydberg states of NO₂. II. Comparing the symmetric stretching and bending vibrations.

Journal of Chemical Physics 120, 2667 (2004)

State-Selective Production of Vibrationally Excited NO₂⁺ by Double-Resonant Photoionization
J. Phys. Chem. A, 2004, 108 (45), pp 9645–9651

VIBRATIONAL AUTOIONIZATION IN POLYATOMIC MOLECULES

Annual Review of Physical Chemistry Vol. 56, 281-308 (2005)

Renner–Teller interactions in the vibrational autoionization of polyatomic molecules

Journal of Chemical Physics 129, 164310 (2008)

As well as more recent references:

Recoil frame photoemission in multiphoton ionization of small polyatomic molecules: photodynamics of NO₂ probed by 400 nm fs pulses

JOURNAL OF PHYSICS B-ATOMIC MOLECULAR AND OPTICAL PHYSICS Volume: 47 Issue: 12 Special Issue: SI Article Number: 124024 Published: JUN 28 2014

Excited state wavepacket dynamics in NO₂ probed by strong-field ionization

JOURNAL OF CHEMICAL PHYSICS Volume: 147 Issue: 5 Article Number: 054305 Published: AUG 7 2017

Spectral dependence of photoemission in multiphoton ionization of NO₂ by femtosecond pulses in the 375-430 nm range

PHYSICAL CHEMISTRY CHEMICAL PHYSICS Volume: 19 Issue: 33 Pages: 21996-22007 Published: SEP 7 2017

Reply to Reviewer Comments on Nature Communications Manuscript

New Title: “Coherent 2D electronic mass spectrometry”

Old Title: “Molecular-beam coherent 2D electronic spectroscopy reveals photoionization pathways”

We thank all reviewers for their detailed comments that helped us improve the clarity of the presentation in the revised version of our work. We have addressed all points in detail as listed below, and we have modified the manuscript accordingly, including additional figures. This reply letter is rather long because the reviewers raised a number of interesting points that required correspondingly detailed answers, owing to the complexity of the topic.

Reviewers’ comments in this reply letter are printed in black, our response in blue, and explicit changes to the manuscript in red font. All modifications are also visible in a marked-up version of the manuscript. Whenever page numbers are provided in our reply, they refer to the revised, marked-up version. If not stated otherwise, referenced numbers also refer to the revised, marked-up version.

We hope that with this extensive reply and the associated changes our manuscript can be recommended for publication.

Reviewer #1

The manuscript title “Molecular-beam coherent 2D electronic spectroscopy reveals photoionization pathways” (“molecular beam” should be unhyphenated) indicates a very promising work. Having 2D coherent spectroscopy available in molecular beam experiments would be a seminal step for a broad spectrum of research. Condensed phase multidimensional spectroscopy is a powerful technique but it is suffering because of unambiguosness in the interpretation coming from the often very complex systems which have been studied. This has already led, in particular in Nature and Science papers, to later on revised interpretations e.g. regarding the excitement on quantum effects in biology. Options of having 2D electronic spectroscopy on molecular beams in that respect will be able to answer fundamental questions on electronic processes of molecular systems. Roeding and Brixner have managed to develop a technique for doing this and are presenting first results. This is indeed original new work of high importance and deserves publishing in Nature Communications.

We thank the reviewer for this very kind and positive evaluation of our work.

Specific remarks to the manuscript:

“molecular beams” (title and text): In the Methods section one finds “sample is expanded as an effusive beam into the vacuum chamber, increasing the base pressure in the interaction region to 1×10^{-4} mbar”. If the experiments probes an increased base pressure in the interaction region one would not call this a molecular beam but an experiment in the gas-phase similar to gas cells. This would still be an interesting important step for 2D spectroscopy but different from molecular beam studies and would not justify the title.

The phrase “molecular beam” generally refers to a directed beam of molecules expanded into vacuum and was used in this context in our manuscript. Depending on the experimental conditions, the expansion can lead to a so called ‘effusive beam’ or a ‘supersonic beam’. Both implementations thus in principle fall under the general heading of “molecular beam”. Often, only the latter case is denoted as a ‘molecular beam’ in literature. Nevertheless, effusive beams have also been called „molecular beams“ in the literature, see, e.g., Swennumson and Even, Rev. Sci. Instr. 52, 559 (1981); DeWitt et al., J. Phys. Chem. B 110, 6705 (2006); Jankunas and Osterwalder, Ann. Rev. Phys. Chem. 66, 241 (2015), and others. This in principle justifies our usage of the term.

However, in order to avoid confusion, we accept the reviewer’s position that the phrase „molecular beam“ might be considered inaccurate by some readers. We had used it to emphasize that our experiment is quite different from experiments using (static) gas cells. In particular, our arrangement allows us to use time-of-flight mass spectrometry as a new detection method for coherent 2D electronic spectroscopy, which is not available in static gas cells as, e.g., applied in the work by Warren and coworkers using fluorescence detection [Ref. 33]. Also, molecular beam technology allows one to investigate a much broader range of substances. Furthermore, upgrading the machine with a specific preparation chamber (construction currently underway in our laboratory) will facilitate a supersonic molecular beam in an otherwise identical experimental setup.

Following the reviewer’s request, and in order to stress the novel aspect of the method more clearly, which is the detection of mass spectra, we changed the title of our manuscript accordingly to **“Coherent 2D electronic mass spectrometry”**.

Within the text, in order to specify the current status of technique in more detail, we changed the phrase “molecular beam” into **“effusive molecular beam”** whenever it is related to our technique, at the following locations: on page 1 in the abstract, on page 2 in “... combining for the first time 2D spectroscopy with **effusive molecular beams...**”, and on page 5 in “... (of **effusive molecular beams**)...”.

In the introduction on page 1 referring to existing literature, we retain the phrase “molecular beam” in the context of its more general meaning.

Section “Multiphoton ionization 2D spectroscopy”: Here, they demonstrate their method and that they can obtain 2D maps specific to a detection of the parent or fragment ion of pas-phase samples. This is new and original; the data are on multiphoton processes which hampers an easy interpretation. Hence, there is no interpretation of the maps for these results in terms of new insight of the system specific to the 2D method.

The reviewer raises an important point that apparently was not made sufficiently clear in the original version of the manuscript: In fact, it is precisely in this regime of multiphoton processes where our approach delivers new insight not available with established literature methods, in the following way: It is a general challenge in the interpretation of time-resolved (multiphoton) mass spectrometry experiments to precisely assign excitation pathways to the observed signals. More specifically, analyzing fragmentation patterns from molecular dissociation and ionization is complicated by the fact that a particular ion-fragment signal might arise either 1) from dissociation on a neutral excited-state potential-energy surface followed by ionization (“ionization of a dissociation product”), or 2) from ionization followed

by dissociation in the ionic manifold (“dissociative ionization”). In reality, both contributions might contribute and overlap. In the words of the reviewer, this indeed generally “hampers an easy interpretation”.

Our 2D mass spectrometry method offers more insight, however, even in this multiphoton regime. For example, in our NO_2^+ case, one might speculate whether both the NO_2^+ and the NO^+ species originate from the same initial excitation pathway, i.e., an ionization to NO_2^+ which is then followed by dissociation toward NO^+ from the same intermediate state. In conventional pump–probe mass spectrometry one could not assess the adequacy of such a hypothesis as argued above. In our 2D experiments, however, we see distinctly different 2D spectra leading to the two products. Thus we conclude that they originate from different intermediate species. This is our interpretation of the results “in terms of new insight of the system specific to the 2D method” as the reviewer requests.

We have modified the discussion to emphasize this aspect more clearly by introducing the following paragraph on page 4:

“One might speculate whether both the NO_2^+ and the NO^+ species originate from the same initial excitation pathway, i.e., an ionization to NO_2^+ which is then followed by dissociation toward NO^+ from the same intermediate state. In conventional pump–probe mass spectrometry one could not assess the adequacy of such a hypothesis, as a particular ion-fragment signal might arise from dissociation on a neutral excited-state potential-energy surface followed by ionization (“ionization of a dissociation product”), or from ionization followed by dissociation in the ionic manifold (“dissociative ionization”). 2D mass spectrometry offers more insight, even in this multiphoton regime.”

Section “Transient 2D spectroscopy”: Here the system is pumped by an extra UV pulse at a fixed delay time; no transients are recorded. The results are interpreted as Fano couplings to continuum states. Model calculation are performed and compared to the results. This is indeed a very interesting case where information is obtained specifically to the 2D method. Again, multiphoton processes and overlapping contributions have to be interpreted. The number of involved photons is nicely assigned in power-dependent measurements and the phase-cycling method can select specific interaction paths, demonstrating the advantages of the introduced technique. I have some doubts on the validity of the interpretation in comparison to the modeled distributions because of the low signal to noise ratio. Do the data, shown in Fig. S3a and S5a, represent the same data set? The main features in the center seem to be identical but the noise only in the outer regions appears to be less in S5 (?).

The experimental data shown in Figs. S3(a) and S5(a) as well as in Fig. 3(a) of the main text are of 8th order in the number of interactions, comprising the interaction with two UV photons and two visible photons that lead to a very weak signal. In the experiment, this signal is additionally obscured by background ions arising solely from the interaction with the UV laser pulse. In order to extract the essential features of the transient signal, we apply a filtering procedure as described at the beginning of Section 3 of Supplementary Information with the following sentences:

„Figure S3(a) displays the transient rephasing 2D spectrum of the NO_2^+ parent ion at a population time $T = 30$ fs. Despite the somewhat noisy appearance, the peak along the

diagonal can be clearly identified. In order to aid identification of essential peak features we apply a Gaussian filter to the raw rephasing time-domain data with a FWHM of 1.4 time pixels (corresponding to 14 fs), thereby removing high-frequency contributions originating from experimental noise while retaining the essential peak features. This yields the rephasing 2D spectrum as shown in Fig. S5(a) and in Fig. 3 of the main text. Such a procedure is justified in our case as the measurement has been performed in the rotating frame, shifting all frequencies to the origin of frequency space and allowing to remove high frequency components originating from experimental noise.”

For me, just based on Fig. S3, I would not dare to do an interpretation in comparison with the theory. The $q=10$ model plot has in total only very weak negative intensity – the experiment is dominated by negative values. The specific streaks in the imaginary part cannot be identified. The resolution in the direction of the diagonal appears to be very low which is not plausible. I would expect that a stronger dissipation (lower Γ value) could explain this. Here no comparison is shown to justify the chosen value and the interpretation of low dissipation.

We apologize for the lack of information on the influence of the Γ factor in the Fano model. We have now extended the comparison with the model and present additional figures containing new data. The dissipation indeed has an influence on the shape of the peak. In order to justify our conclusions, we now calculate the 2D lineshape for different values of Γ and q and plot them arranged in a matrix-like structure for easy comparison, thereby replacing Fig. 3(d) and Fig. S4 of the previously submitted manuscript. As can be seen from the plots, a stronger dissipation (smaller Γ) increases the overall linewidths (diagonal and anti-diagonal), whereas q influences the overall symmetry of the peak shape and leads to a completely positive signal in the case of $q < 10$ values.

In order to extend the discussion, we added the following sentence on page 7 of the main text: “Additionally, for increasing Γ (top to bottom), the dissipation is weaker and the linewidth decreases.”

Furthermore, we provide more details on page 5 of Supplementary Information: “A large q parameter denotes weak coupling and a dominating transition to discrete auto-ionizing states. In contrast, small q denotes either a strong direct transition from the 3d Rydberg states to the ionic continuum or strong coupling via the auto-ionizing states.” And we changed on page 6 of SI “We have chosen $\Gamma = 1$ and $q = 160 \text{ cm}^{-1}$ for qualitative comparison” to “We have chosen $\gamma_e = 160 \text{ cm}^{-1}$ for qualitative comparison of the influence of Γ and q on the 2D lineshape in Fig. S4.”

Concerning the reviewer’s comment on “specific streaks in the imaginary part”, we agree that the agreement between experiment and model is not perfect. Indeed, it cannot be perfect because the analytic Fano model has been developed for a single discrete state, whereas we know that in the case of NO_2 multiple levels would have to be considered. However, such a model is not yet available in the literature. Thus we hope our work will trigger future studies in theoretical groups addressing this fascinating question.

Indeed, the existing discrepancy at the current status of experimental and theoretical data, which are both imperfect, may be interpreted as a strong point for our novel technique,

rather than a drawback, in the following sense: Already with the present first implementation, one can see that the method offers a sensitive testing ground for high-level electronic structure and dynamics calculations because with an imperfect model it is not possible to obtain a perfect fit.

We now include this argument in an extended discussion that compares the current situation to that of the early works of conventional 2D spectroscopy, on page 7 of the revised manuscript:

“To the best of our knowledge, this case has not yet been elaborated in the literature with respect to lineshapes in coherent 2D spectroscopy and remains an interesting direction for future studies of bound-continuum transitions in order to get quantitative fits through improved modeling and increased experimental signal-to-noise ratios. The situation is comparable to that in the beginnings of conventional 2D electronic spectroscopy of discrete states 15 years ago. For the first multichromophore measurements, agreements between experiment and theory were mainly qualitative [19]. Even so, it was clear already at the time that 2D electronic spectroscopy is a powerful method offering new insights. With the present first implementation of coherent 2D electronic mass spectrometry, we have reached now a comparable first step for the 2D spectroscopy of unbound (ionic continuum), rather than bound (discrete), states and one can see that the method offers a sensitive testing ground for high-level electronic structure and dynamics calculations.”

Concerning the additional reviewer’s comment on the spectral resolution, see the clarification in the following item of our reply.

On the other hand, in the direction perpendicular to the diagonal, the spectral resolution appears to be very high (change from +1 to -1 from one pixel to the next and back). From that I conclude the experimental resolution to be high (according to the 200fs scan width).

The frequency resolution of the 2D spectra shown in our manuscript is determined by the maximum delay in the time domain as the reviewer notes. The experimental parameters yield a frequency resolution of 159 cm^{-1} along both directions as quoted on page 5 of the main text. In order to make the spectral resolution clear at first sight, we have decided to plot all 2D data in a pixelated version, wherein the size of one pixel corresponds to the spectral resolution.

We have added a sentence on page 5 to point this out: “All 2D spectra are shown in a pixelated version, wherein the size of one pixel corresponds to the spectral resolution (159 cm^{-1}).”

For the signal shown in Fig. S3, this frequency resolution is sufficient to resolve the peak structure along the diagonal, i.e., the diagonal elongation is a signature of the molecular response and not limited by the method. However, in the case of the anti-diagonal direction the experimental resolution is not sufficient to fully resolve the peak shape. In particular, we cannot observe the gradual change of sign that is intrinsic to the response function and find instead an abrupt change consistent with the resolution. We attribute the origin for the apparently quite different 2D peak widths in the diagonal versus anti-diagonal directions to the superposition of several transitions underneath one peak as stated in the main text and as also discussed in the previous item of our response above.

We added an additional explanation at the end of the first paragraph on page 6 of the main text to make this clearer: “The peaks line up along the diagonal, with a very narrow width across the anti-diagonal as expected from the small linewidth due to the isolated conditions in the gas-phase sample. This leads to an effective peak shape that is much broader in the diagonal than in the anti-diagonal direction.”

Section “transient absorption spectroscopy”: I would not call the results of this part a “new type of ... spectroscopy”. The technique represents a one-dimensional Fourier transform spectroscopy which goes in many variants 70 years back to the work of Ramsey. Characterization of electronically excited states with ion detection in the gas-phase has already been done in the 90s (e.g. Chem.Phys.Lett. 233 (1995) 491-499).

We followed the recommendation, added the suggested and some other references, and rephrased the introductory sentence of this section on page 7, pointing out the relation of the earlier work and our new implementation here. The main novel aspect is that we observe *transient* changes in the absorption spectrum. In that sense our work goes beyond the earlier work in which the one-dimensional Fourier transformation contained the full information from the experiment because there was only one time coordinate. In our case, this transform is performed for varying waiting times, and thus we obtain transient, as opposed to static, spectral information:

“In the next step, we employ the versatility of the phase-cycling approach and realize pump–probe-like “transient absorption” spectroscopy on molecular beams without the need for measuring absorption changes directly. Obtaining spectral information from time-dependent measurements in molecular beams has been demonstrated in the 1990s [3,4,58,59]. In those cases, the spectral signature of the transition to the excited state was obtained revealing, for example, vibrational level spacings. In the present work, we show how *time-dependent* absorption can be extracted, i.e., in relation to the mentioned pioneering work we add an additional time coordinate and observe how spectral properties evolve.”

In a rate equation model the pump-probe data are fitted. Here one should be careful fitting the 2 transients with 5 parameters, and even for the NO⁺ signal not getting good agreement (increasing slope around 0). The authors should make clear that the numbers and error margins are justified.

First of all, we point out that the details of the fitting procedure of the pump–probe data are not essential for the introduction of the 2D method and neither for the interpretation of the 2D data. This part of the work is mainly incorporated to facilitate a comparison with what is known in the literature and to offer some validation of the 2D data via projection onto an existing technique. Anyway, of course, the pump–probe data has to be treated diligently.

Let us hence consider the particular case of NO₂ which is known to have very complex dynamics especially in the first excited states (which can be reached by pumping with visible light, i.e., at negative delay times). In order to motivate the chosen model to fit the data and to explain the origin of the error margins, we added the following paragraph to Chapter 4 on page 6-7 of Supplementary Information:

“The model used to fit the pump—probe data is chosen in such a way that it results in a correct fit over the full time-domain range with the smallest possible number of parameters. The main intention was to describe the data for positive delay times as accurately as possible, as the transient 2D experiment is also performed at a positive delay between the UV and the visible pulses. We varied the number of parameters used in the fitting model to describe the dynamics at positive and negative delay times and only obtained a converging fit for the presented configuration comprising in total six decay constants (two at negative and four at positive delay times).

The parameter uncertainties result from the square root of the diagonal entries of the covariance matrix C that is calculated via

$$C = [J'xJ]^{-1} \times \sigma,$$

with the mean-squared error σ of the fit and the Jacobian J and the transposed matrix J' that is returned by the applied Levenberg-Marquardt solver.”

In general, the manuscript is very well written, and, in particular, related work is introduced and referenced. The certainly complicated matter of multidimensional spectroscopy of higher order processes is presented in a comprehensible and well-structured way. I would suggest reducing the redundancy of figures.

We thank the reviewer once more for the positive evaluation and followed the suggestion to reduce the redundancy of figures by removing Fig. S6 of Supplementary Information which contained the same information as Fig. 5 of the main text. We could not find further redundant figures.

In conclusion, the manuscript presents a milestone in coherent spectroscopy but needs revisions due to the remarks above. In particular, the interpretation in comparison with the Fano model has to be convincing because this is the scientific achievement obtained by the new method.

We hope that with the modified discussion in the section on transient 2D spectroscopy regarding the origin of the 2D peak shape, the influence of the Fano Γ parameter on the 2D linewidth, and with extended figures for the 2D Fano model, the comparison between theory and experiment and what can potentially be learned from it have become clear now. In this section on transient 2D spectroscopy, we additionally emphasize the potential outreach of our new method for the investigation of bound-continuum transitions.

Reviewer #2

In this manuscript, the authors present a new technique that combines 2D electronic spectroscopy with time-of-flight mass detection. This new technique measures the photogenerated ions as a function of time delays and relative phases of the excitation pulse sequence. The ion yield in the resulting mass spectrum serves as a reporter of the final-state population. By directing both UV and visible beams to the sample and selecting different excitation sequences, the authors are able to perform multi-photon ionization 2D spectroscopy, transient 2D spectroscopy, and transient absorption spectroscopy with the same optical setup. The authors further prove the capability of their new technique by investigating the photodynamics of NO₂. With multiphoton 2D spectroscopy, the authors reveal that the parent ion NO₂⁺ and the fragment ion NO⁺ originate from different intermediate species. With transient absorption spectroscopy, the authors identify transitions from 3d Rydberg states of NO₂ to auto-ionizing states.

The idea of combining 2D electronic spectroscopy is novel and brilliant. I realize that the implementation of this technique is difficult, and I am impressed by the data presented in this manuscript. This new technique renders possibility to detailed studies on the role of the environment and coupling between different states in primary steps of photochemistry. This manuscript will inspire research on both nonlinear spectroscopy and gas phase dynamics and is certainly worthy of publication in Nat. Commun. However, there are a few major issues that the authors need to address regarding to details of their experimental procedure and data interpretation. I therefore recommend publishing this manuscript after revision.

We thank the referee for this very positive evaluation of the manuscript and address all further comments below.

Major comment:

1. The energy requirements to generate NO₂⁺ must be well justified and explained. In the power dependence study presented in SI, the authors claim that the two-color pump-probe experiment requires 2 UV photons and 1 photon of the visible pulse to generate NO₂⁺. But they claim that the 2D experiment requires 4 photons from the visible pulse to generate NO₂⁺. The UV beam in this experiment is centered at 267 nm with pulse duration of 130 fs. The visible beam is centered at 540nm with FWHM of 34 nm and pulse duration of 17 fs. My question is: If 4 visible photons have enough energy to populate the system into auto-ionization states, why 2 UV photons cannot?

This is a very important question and we accordingly extended the discussion in the revised version of our manuscript. The experiments presented in our manuscript are not sensitive to any UV-only photoionization process by virtue of how we carry out the measurement, i.e., we only detect a change in the ion signal originating from the interaction with the UV *and* the visible pulse. As described on page 11 of the main text, we experimentally detect an ion signal originating solely from the interaction with the UV pulse by blocking the visible probe pulse at each time delay. During the course of data analysis, this background signal is subtracted in order to obtain the pure pump–probe signal arising from the interaction of both pulses.

In order to gain more insight into the process of UV photoionization and to answer the specific question of the referee, we now include new data in which a power-dependent UV-only ion signal was acquired. In the revised version of our manuscript we show the data of this measurement in new Fig. S1a of Supplementary Information and further added the following paragraph on page 1:

“Figure S1(a) displays the ion yield power dependency using the UV pulse only. The linear fit yields a slope of 2.77 for the parent ion, indicating that ionization with the UV pulse occurs by two competing processes comprising two or three photons, respectively. The fragment power dependency yields a slope of 2.70 meaning that it requires a similar amount of energy to generate the fragment ion via the ionization of a dissociation product as discussed in the main text.”

We further add the following paragraph on page 5 of the main text in order to clarify the effect of the UV pump pulse on the transient 2D spectroscopy experiment:

“The UV pulse itself is intense enough to produce a constant level of background ions via (resonant) multiphoton ionization as detailed in Supplementary Information. Using phase cycling in the visible probe-pulse sequence and analyzing the data accordingly, transient 2D spectroscopy is, however, only sensitive to signals that arise from the interaction of both, the UV as well as the visible probe pulses and hence automatically discards ion contributions originating only from the UV pulse. Ions generated only by the visible pulses were not detected due to a lower laser intensity in this experiment.”

The authors claim that the vertical ionization potential of NO₂ of the equilibrium geometry of the ground state is 11.23 eV, but the equilibrium geometry of the molecule at excited state surface can change from a bent configuration to a linear configuration, which is identical to ionization energy of 9.586 eV and can be assessed by 4 visible photons. Then how quick is the change of geometry? The 2D spectrum of NO₂ are taken at T = 100 fs. If this change of geometry happens within 100 fs, will the 2 UV photons do the same thing because the UV beam has pulse duration of 130 fs?

This question is also highly relevant for the discussion of our findings. The Rydberg states excited by the absorption of two UV photons / four visible photons correspond to a linear geometry of the molecule, in contrast to the ground-state configuration with a bending angle of $\alpha = 134.25^\circ$. Thus, excitation into Rydberg states is accompanied by a change in geometry and the question remains at which level of excitation the molecular dynamics proceed.

In time-resolved imaging experiments using 400 nm pump pulses with a pulse duration of 150 fs, Vredenburg et al. [42] observe excitation into Rydberg states by the absorption of 3 pump photons (equivalent in energy to 2 UV photons or 4 visible photons in our experiment). Excitation proceeds via an intermediate resonant state at the level of the first photon. They speculate that within their relatively long pulse duration the molecular geometry changes to a linear geometry, enabling the absorption of two additional photons into the linear Rydberg states.

For our experiment, a similar reasoning can be applied, as there are possible resonant intermediate states at each energy level that can be accessed by an increasing number of

visible or UV photons (see also **new Fig. S2(d)** in Supplementary Information with a list of accessible states). With a pulse duration of 130 fs, the UV pulses seem long enough that the change of geometry happens within the laser pulse.

Even though the visible pulses are considerably shorter than 150 fs, the dynamics can proceed during the waiting time of $T = 100$ fs. Furthermore, this process could in principle be followed by scanning T which was not done here due to a considerable increase in measurement time.

We added the following paragraph on page 5 of the main text:

“Ionization using four photons is accompanied with a change from bent to linear geometry at a resonant intermediate level [42] that can possibly proceed during the waiting time of $T = 100$ fs. This process could in principle be followed by scanning T which was not done here due to a considerable increase in measurement time.”

2. In the multiphoton 2D spectroscopy part, the authors state that the ionization of NO_2^+ requires 4 visible photons and is in line with previous (1+2+1) multiphoton-ionization process. They also claim that it requires 5 visible photons to generate NO^+ . But what is (are) the detailed ionization pathway(s) in this experiment, (2+2+1) or others? To my knowledge, there are multiple ionization pathways to generate NO^+ (J. Chem. Phys. 75, 2643–2651 (1981); Annu. Rep. Prog. Chem., Sect. C: Phys. Chem. 106, 274–304 (2010)). Are the NO^+ coming from dissociation and ionization (NO_2 to NO to NO^+ , as they mentioned in the manuscript, Section multiphoton ionization spectroscopy) or ion-pair state and dissociative ionization (NO_2 to ion-pair state to NO^++O^- , as they mentioned in SI, Section 4)?

This question is very similar to Reviewer 1’s comment on “Multiphoton ionization 2D spectroscopy”. There, on page 2 of this reply letter, we discuss the challenges that arise when trying to identify the exact process leading to an ionic fragment. We repeat the main points of the argument here. In the general case, various ionization and dissociation pathways may indeed occur and cannot easily (or at all) be distinguished with conventional methods. We argue, however, that our new 2D method offers more information and repeat the reasoning from page 2 of this reply letter here:

Our 2D mass spectrometry method offers more insight, however, even in this multiphoton regime. For example, in our NO_2 case, one might speculate whether both the NO_2^+ and the NO^+ species originate from the same initial excitation pathway, i.e., an ionization to NO_2^+ which is then followed by dissociation toward NO^+ from the same intermediate state. In conventional pump–probe mass spectrometry one could not assess the adequacy of such a hypothesis as argued above. In our 2D experiments, however, we see distinctly different 2D spectra leading to the two products. Thus we conclude that they originate from different intermediate species.

We have modified the discussion to emphasize this aspect more clearly by introducing the following paragraph on page 4 of the main text:

“One might speculate whether both the NO_2^+ and the NO^+ species originate from the same initial excitation pathway, i.e., an ionization to NO_2^+ which is then followed by dissociation toward NO^+ from the same intermediate state. In conventional pump–probe mass

spectrometry one could not assess the adequacy of such a hypothesis, as a particular ion-fragment signal might arise from dissociation on a neutral excited-state potential-energy surface followed by ionization (“ionization of a dissociation product”), or from ionization followed by dissociation in the ionic manifold (“dissociative ionization”). 2D mass spectrometry offers more insight, even in this multiphoton regime.”

Thus we obtain more insight with 2D mass spectrometry as compared to established methods. The full separation into the various pathways that the reviewer desires to know would require a more complex phase-cycling scheme and an even larger number of pulses. For a discussion of these extended schemes, please refer to our reply to “Minor comment 4” below on page 15 of this reply letter.

3. The authors draw an energy diagram that is relevant to the two-color pump probe experiment in Fig. 5b. However, there are no energy diagrams for the multi-photon 2D measurements or transient 2D measurements in the current manuscript. Different excitation sequences and beams will access different electronic states. Although there are multiple high energy states in NO₂, the electronic structure of NO₂ is well studied in previous literatures (Annu. Rep. Prog. Chem., Sect. C: Phys. Chem. 106, 274–304 (2010); J. Chem. Phys. 115, 10394-10403). The authors should provide energy diagrams for the other two measurements, which can help the data interpretation.

In the original version of our manuscript, we have chosen not to draw energy diagrams for the experiments other than pump—probe in order to avoid redundancy of figures. But we have now included an assignment of reachable states for the various experiments in modified Figure 5(b).

For an explanation of multiphoton processes and the usage of labels in the Feynman diagrams, please refer to our answer to the following comment 4.

4. The Feynman diagrams in Fig S2.c are misleading. For example, in the first diagram on the top row of Fig S2.c, the first two field interactions lead to the transition from $|g\rangle$ to $|e1\rangle$. But in the second diagram on the top row of Fig S2.c, the first field interaction leads to the transition from $\langle g|$ to $\langle e1|$. The authors should draw their Feynman diagrams according to the electronic structure of NO₂ because the states have to be accessible with their laser pulses. The nomenclature e1, e2, f should be replaced with actual states of NO₂.

The labelling of levels in Feynman diagrams is a point of frequent confusion. We chose the labels in Supplementary Information in a generic fashion. This follows in style the language introduced by Shaul Mukamel [51] in his generalization of nonlinear spectroscopy in terms of the response-function formalism. First, generic labels are employed that are “place holders” for the states reached after individual electric-field interactions (Chap. 7 in Ref. [51]). Then, in a second step and depending on the particular model, those labels are expanded into the “real” states present in the system. Thus, also in our description, equal labels contained within two different Feynman diagrams do not correspond exactly to the same state because they are simply place holders that in a real calculation need to be summed over all accessible states. If one were to plot those diagrams with all the real states, this would result in a confusing number of Feynman diagrams, especially in the case of NO₂, since many Feynman diagrams with different intermediate states are conceivable.

Thus, there are two reasons why we did not follow directly the reviewer's request that the "nomenclature e1, e2, f should be replaced with actual states of NO₂": 1) The numbers of diagrams would get prohibitively excessive and would thus not add to the clarity of the presentation, and 2) such labelling would seem to reduce inappropriately the generality of the method itself which we introduced here.

Nevertheless, we have found a way to incorporate the request into the manuscript and at the same time keeping the generality: We added Figure part S2(d) and write on page 3 of SI:

"Figure S2(d) shows possible assignments of the general energy-level "place holders" of the Feynman diagrams of Fig. S2(c) to specific states in the example of NO₂."

For the full description, one then needs to calculate the Feynman diagram contributions for all combinations of accessible levels.

5. Similarly, for the Feynman diagrams with 10th-order contributions that give rise to NO+ signals, different ionization pathways will give different Feynman diagrams and it is important to draw the dissociation process if it happens during the waiting time of certain pathways.

We are not sure if we fully understand this comment, but we interpret it in the following sense: The reviewer seems to ask whether different Feynman diagrams result depending on whether dissociation occurs within the pulse sequence or not.

The question is connected to comment 4 and its answer above. Since the reactant, NO₂, and one of its detected products, NO⁺, are distinct chemical species, each of which has distinct (and different) electronic and vibrational levels, one is tempted to request the usage of labels pertaining to those respective levels within the Feynman diagrams. Then one would have to distinguish at which point of time and during which interaction in the Feynman diagram the dissociation would take place. This would lead to Feynman diagrams employing different labels for states at various steps, i.e., some of the labels referring to NO₂/ NO₂⁺ states and some of the labels to NO/NO⁺ states. We are not sure if such an approach will lead to consistent results in the end because different electronic level descriptions are mixed up and it is not clear what type of wavefunction (in an actual calculation) one would have to enter to connect the two sets of levels correctly. Even if that is possible, the visualization of diagrams would be complicated tremendously, and in our opinion this would lead to an undesirable loss of clarity compared to the already complicated, more generic, diagrams.

Anyway, we believe a different viewpoint is more appropriate. Since we deal with a unimolecular dissociation process, the complete chemical system of NO₂, NO₂⁺, NO, and NO⁺, is describable in one unifying potential energy landscape, where the dissociation product is simply characterized as the limit of the surface when one of the bond-length coordinates goes to infinity. Thus, the full dynamics are describable on one potential-energy landscape. It is not necessary (and indeed counterproductive) to label the states of NO/NO⁺ separately because anyway the coherent dynamics between the "species" need to be conserved. Thus, it is better to keep the "generic" labels also mentioned in comment 4 above. Assigning these labels in the new Figure S2(d) to the states of NO₂ makes sense because this is the starting point. Of course one has to keep in mind implicitly or, if quantitative treatment is desired, model explicitly the full wavepacket dynamics during

interaction with the pulse sequence. Dissociation then can be said to have occurred when during that wavepacket propagation the bond stretches above a certain threshold.

The literature on theory of 2D spectroscopy has mainly focused on the response-function treatment summarized by Mukamel that can also take into account vibrational levels. However, explicit wavefunction-based calculations have also been developed [56,57] that allow explicit propagation of quantum dynamics that may be generalized toward dissociative processes as well as ionization to continuum states. Such approaches could thus in the future be employed to deal with systems and questions introduced in the present work, and we hope that our publication would stimulate such research. At present, however, such a treatment would go way beyond the scope of the present work that introduced the new spectroscopy method itself.

We added a corresponding comment on page 7 at the end of the section on transient 2D spectroscopy:

Future research on the detailed theoretical treatment of 2D spectra of dissociative processes as well as ionization to continuum states would benefit from, e.g., the generalization of wavefunction-based quantum dynamics calculations [56,57] and a consistent approach to visualize response functions resulting from dissociation processes within Feynman diagrams.

6. On page 7 of the main manuscript, second to the last line, the authors state that they scan τ to achieve the time delay between the UV and visible beams. But in the Method section (Page 11), they claim that they scan Δ for the pump-probe measurements. Which experimental parameter is actually scanned in the experiment?

As indicated in Fig. 1, we perform three different experiments (labelled a, b, c) that are discussed throughout the text. The section on page 7 refers to the transient 2D spectroscopy experiment displayed in Fig. 1(b), where we vary the time delays τ and t whereas Δ is kept at a fixed delay. During the regular pump-probe experiment [Fig. 1(c)], we only vary Δ .

On page 7 of the main text, we compare the transient-absorption contribution obtained from the experiment using four probe pulses and phase cycling [Fig. 1(b)] with a regular pump-probe experiment [Fig. 1(c)]. For the selected signal contribution, scanning the delay τ in Fig. 1(b) corresponds to scanning the delay Δ in the pump-probe experiment of Fig. 1(c), as the molecular system is in the same population state in both configurations.

In order to clarify the properties of the different experiments we modify and extend the description on page 9:

“For direct comparison of pump-probe data obtained from the visible four-pulse sequence [configuration (b) of Fig. 1] via phase cycling with a conventionally obtained pump-probe data set (without frequency resolution) using just one single pump and one single (ionization) probe pulse [configuration (c) of Fig. 1], the time-domain data [Fig. 4(b)] is integrated over t , providing the green data points in Fig. 4(d). For the selected signal contribution, scanning the delay τ in Fig. 1(b) corresponds to scanning the delay Δ in the pump-probe experiment of Fig. 1(c), as the molecular system is in the same population state in both configurations.”

7. In the pump-probe measurement, the authors attribute NO^+ signal at short delay times to ion pair dissociation (NO^+ and O^-) after (2 UV+ 1 Vis) excitation. They have also attributed the long-lived NO^+ signal to neutral NO_2 dissociation from 3d Rydberg states after 2 UV photon excitation and formation of excited NO in the $\text{A}^2\Sigma^+$ state followed by subsequent ionization. But is 1 visible photon enough to ionize NO ? Or, this process requires multiple photons? From Fig 5b, it seems like 1 visible photon cannot provide enough energy.

It is indeed interesting to ask about the order of the ionization process generating the NO^+ fragment at large delay times. In fact, we actually performed this measurement, but since the signal is small a reliable fit cannot be achieved, which is why we do not show the probe power dependency for the transient NO^+ signal in Supplementary Information. We added the following paragraph on page 1 of Supplementary Information to address this point:

“As can be inferred from the transients shown in Fig. 5 of the main text, the fragment ion signal is much weaker compared to the parent ion signal at positive delay time. Thus, it is stronger affected by experimental noise, which especially in low-power conditions would obscure the power dependency. Therefore, we did not obtain the number of photons contributing to the NO^+ transient ion signal, as the range of the signal was much too small to yield a reliable fit.”

Minor comments:

1. The time delay label in Fig 1 beam configuration c needs to be consistent with the Feynman diagram in Fig 4a.

We apologize for the lack of precision regarding the assignment of the displayed results to the experiments represented by the pulse sequences in Fig. 1. Figure 1(c) represents the pump-probe experiment with the corresponding data shown in Fig. 5. In contrast, the Feynman diagram in Fig. 4(a) actually originates from the experiment displayed in Fig. 1(b). In this specific experiment, phase cycling allows us to select contributions that originate from only two interactions with the visible pulses, and this is shown in the Feynman diagram in Fig. 4(a). Thus our labelling was correct. Our answer to comment 6 above aims at resolving this seeming inconsistency, and we changed the manuscript accordingly:

“For direct comparison of pump-probe data obtained from the visible four-pulse sequence [configuration (b) of Fig. 1] via phase cycling with a conventionally obtained pump-probe data set (without frequency resolution) using just one single pump and one single (ionization) probe pulse [configuration (c) of Fig. 1], the time-domain data [Fig. 4(b)] is integrated over t , providing the green data points in Fig. 4(d). For the selected signal contribution, scanning the delay τ in Fig. 1(b) corresponds to scanning the delay Δ in the pump-probe experiment of Fig. 1(c), as the molecular system is in the same population state in both configurations.”

2. The energy axis needs to be added to Fig. 5b.

We added the energy axis accordingly in a modified Fig. 5b.

3. There is a typo in Figure S2 caption. “nonlinearity that are contributing to the rephasing fourth-order signal shown in Figs. 5(a) and (c)” should be “ Figs. 2(a) and (c)” instead.

We corrected this typo in the revised manuscript.

4. The authors provide data processing and phase cycling equations for fourth-order signals, yet the signals that contribute to the 2D spectrum are 8th and 10th order signals. I recommend the authors to add a few sentences in the SI, explaining how these higher order signals survive with their current phase-cycling scheme and what higher step phase cycling scheme can in principle discriminate between these signals.

This “minor comment” is rather a “major comment”, at least judging from the amount of complexity this entails and considering that nonlinear spectroscopy has not been treated in a systematic way up to that order in the literature before. This is arguably the most complex of all questions in this reply letter. Nevertheless, we extended the discussion on different experimental approaches for nonlinear spectroscopy in order to illustrate the general features of phase cycling for such high orders.

In the literature, the question of the ideal phase-cycling scheme to extract a certain signal contribution has been addressed in the past for experiments using up to four collinear laser pulses [66]. Generally, all fourth-order signals may suffer from higher-order aliased contributions if those higher-order signals are of comparable magnitude as the fourth-order signals. Thus, in order to address the relation of phase cycling to the particular nonlinear order, we added the following paragraphs at the end of Section 2 in SI on pages 3-4.

“In this experiment, the most economic phase-cycling scheme was chosen that allows us to resolve the oscillation of a single-quantum coherence signal as a function of the relative phase between two consecutive pulses in a four-pulse experiment without aliasing. For higher-order nonlinear processes as shown in Fig. S2(c), double (triple) quantum coherences are excited that oscillate rapidly as a function of the relative phase between two pulses and are thus undersampled by the discrete phase increments chosen in our case. Hence, analogous to an ordinary Fourier transformation, aliased signals with rapid oscillations overlap with signals of lower frequency.

The question remains how these highly nonlinear signals could be separated after all. Three approaches are conceivable. The first possible approach is to refrain from using phase cycling and instead to perform the measurement in the laboratory frame ($\gamma = 1$). In the case of 8th (10th) order contributions, 7 (9) time delays need to be scanned with a temporal resolution of about 1 fs in order to fully resolve all contributions, rendering this an extremely complicated experiment. The second option is to perform the measurement in a fully rotating frame and to use an appropriate phase-cycling scheme to uniquely discriminate all contributions. Aiming to resolve contributions where each of the 8 pulses only interacts once, a 1x3x3x3x3x3x3x3x3 phase-cycling scheme is necessary, similar to the case of four collinear laser pulses.

Generally, an intermediate approach using a partly rotating frame (e.g., $0 < \gamma < 1$) for the 7 (9) time delays and a reduced phase-cycling scheme is also possible, combining both approaches.”

5. The authors may consider changing the title of the manuscript. In the current manuscript, the photoionization pathways are identified by the two-color pump probe measurement, not multiphoton or transient 2D spectroscopy. Although I understand these are difficult measurements, it is still too far from claiming the 2D spectra reveal ionization pathways.

We agree with this suggestion (as well as that of Reviewer 1) and changed the title accordingly to “Coherent 2D electronic mass spectrometry”.

In summary, the concept of combining 2D electronic spectroscopy with mass detection is creative and insightful. I am impressed that the authors successfully performed multiple measurements on NO₂ and present high quality data on high order nonlinear signals. This is a tour de force experiment. This manuscript is certainly interesting to nonlinear spectroscopy community and photochemistry/photophysics community. This paper should absolutely be published in Nature Communications after suitable revision.

We thank this reviewer once again for the very positive evaluation and especially for acknowledging the high quality of the data. We hope that with the extensive changes detailed above the manuscript can now be recommended for publication.

Reviewer #3

The authors present a 2D spectroscopy probe of NO₂ in the gas phase. The work is technically proficient and carefully analyzed. However, this seems to be a technical demonstration of 2D spectroscopy in the gas phase and no new information about NO₂ spectroscopy or dynamics was obtained. Much is known about this molecule and the authors demonstrate no advances compared to this extensive body of existing knowledge. The motivation for 2D gas phase spectroscopy has not been convincingly made: there seems to be no new information obtained from this study.

First, we would claim that a “technical demonstration” of 2D spectroscopy in the gas phase in combination with mass spectrometry is by itself a significant achievement because it opens new directions of research. The significance of such work is also recognized by Reviewer #1 as follows: “Condensed phase multidimensional spectroscopy is a powerful technique but it is suffering because of unambiguity in the interpretation coming from the often very complex systems which have been studied. This has already led, in particular in Nature and Science papers, to later on revised interpretations e.g. regarding the excitement on quantum effects in biology. Options of having 2D electronic spectroscopy on molecular beams in that respect will be able to answer fundamental questions on electronic processes of molecular systems.”

Hence, gas-phase 2D spectroscopy can serve as a complementary tool to condensed-phase techniques investigating transitions/coherences between bound states and the influence of the bath environment on coherent processes. We motivate our study with the desire to investigate not only bound-bound but also, for the first time, bound-continuum transitions and corresponding lineshapes using 2D spectroscopy. Additionally, our technique enables monitoring of unimolecular dissociation processes with the detection of the complete product mass spectrum.

In order to clarify the motivation of the topic, we added the following paragraph to the end of the introductory section on page 2 of the main manuscript:

“Coherent 2D spectroscopy on molecular beams can serve as a complementary tool to condensed-phase techniques resolving otherwise congested 2D spectra of transitions in complex systems due to the narrow linewidths of a gas-phase sample. It further enables the investigation of bound-continuum transitions and corresponding lineshapes by detecting the ionization products.”

We selected the well-known NO₂ deliberately as a benchmark system for introducing our new method precisely because there exists a vast amount of literature on its ultrafast dynamics that our approach can be compared with. Introducing a new method to our mind in general calls for a (largely) known system first without necessarily requiring new insights. Had we chosen to introduce the new method on an unknown system instead we could have claimed many new insights but would not have had any way of validation.

Having said that, we still have obtained some new insight into the photochemistry of NO₂. This was not made sufficiently clear in the original version of our manuscript, and we remedied this in the revised version of the manuscript. A similar point was raised by Reviewer 1 in the comment on the Section “Multiphoton ionization 2D spectroscopy”. We

now discuss how new information is obtained from this study by introducing the following paragraph on page 4 of the main text:

“One might speculate whether both the NO_2^+ and the NO^+ species originate from the same initial excitation pathway, i.e., an ionization to NO_2^+ which is then followed by dissociation toward NO^+ from the same intermediate state. In conventional pump–probe mass spectrometry one could not assess the adequacy of such a hypothesis, as a particular ion-fragment signal might arise from dissociation on a neutral excited-state potential-energy surface followed by ionization (“ionization of a dissociation product”), or from ionization followed by dissociation in the ionic manifold (“dissociative ionization”). 2D mass spectrometry offers more insight, even in this multiphoton regime.”

This paper should be re-submitted to a specialist physical chemistry journal. Even then, were I to referee it for this case, I would still insist upon a proper review of the well-known spectroscopy and dynamics of this molecule (see the partial list below) and a clear demonstration of some new advance in understanding. Otherwise, it seems to be a purely technical demonstration without any clear evidence of the need for or advantage of this type of spectroscopy for gas phase dynamics.

The authors seem unaware of extensive, detailed work on NO_2 by S.T. Pratt (Argonne):

Mode-dependent vibrational autoionization of NO_2 .
Journal of Chemical Physics 119, 10146 (2003)

Mode dependent vibrational autoionization of Rydberg states of NO_2 . II. Comparing the symmetric stretching and bending vibrations.
Journal of Chemical Physics 120, 2667 (2004)

State-Selective Production of Vibrationally Excited NO_2^+ by Double-Resonant Photoionization
J. Phys. Chem. A, 2004, 108 (45), pp 9645–9651

VIBRATIONAL AUTOIONIZATION IN POLYATOMIC MOLECULES
Annual Review of Physical Chemistry Vol. 56, 281-308 (2005)

Renner–Teller interactions in the vibrational autoionization of polyatomic molecules
Journal of Chemical Physics 129, 164310 (2008)

As well as more recent references:

Recoil frame photoemission in multiphoton ionization of small polyatomic molecules: photodynamics of NO_2 probed by 400 nm fs pulses
JOURNAL OF PHYSICS B-ATOMIC MOLECULAR AND OPTICAL PHYSICS Volume: 47 Issue: 12
Special Issue: SI Article Number: 124024 Published: JUN 28 2014

Excited state wavepacket dynamics in NO_2 probed by strong-field ionization
JOURNAL OF CHEMICAL PHYSICS Volume: 147 Issue: 5 Article Number: 054305 Published:
AUG 7 2017

Spectral dependence of photoemission in multiphoton ionization of NO₂ by femtosecond pulses in the 375-430 nm range

PHYSICAL CHEMISTRY CHEMICAL PHYSICS Volume: 19 Issue: 33 Pages: 21996-22007

Published: SEP 7 2017

We apologize for not considering the suggested references in the main text during our first submission. We are fully aware of the extensive work done in the past on NO₂. Indeed, as argued in our reply to the previous point above, the wealth of knowledge on the system was one of the reasons to choose it as a model system. On the other hand, a large body of literature necessarily entails a painful selection on the numbers of references, simply for space limitations. We had strived to strike a balance between the conflicting desires to provide a comprehensive account on the one hand and a compact manuscript with limited references on the other hand. In the initial submitted version, we achieved this by concentrating on time-resolved experiments and review articles, and Reviewer 1 at least finds that this was a successful attempt by stating that "...related work is introduced and referenced."

Nevertheless, we gladly follow the request of the reviewer and added the following paragraph and references on page 5 of the main text:

Multiphoton ionization can proceed via autoionization [43,44] that depends on the vibrational quantum numbers and mainly proceeds via the symmetric stretching mode in NO₂ [45, 46], which was demonstrated in the mode-selective production of NO₂⁺ [47].

We further added those new references to the existing literature on page 8 of Supplementary Information.

REVIEWERS' COMMENTS:

Reviewer #1 (Remarks to the Author):

Besides the general reply we were explicitly asked to comment on the criticisms of Reviewer#3 and the corresponding reply. The main point was that "a clear demonstration of some new advance in understanding" the studied system was missing. My own assessment in the first round went also in that direction: the scientific achievement (only) is the coupling to continuum states and the comparison with the Fano model; these, however, have not been presented convincingly. The long and extensive reply of the authors has clarified the situation on, I would say, all the raised issues by the reviewers and furthermore demonstrated that the presented work is on the highest scientific level and the authors have thoroughly analyze their data. However, the new scientific insight remains sober. Connected to the Fano model the authors clearly say that the work only demonstrates what "potentially can be learned from it" and, including all sections, I still cannot distil a clear answer on what has been learned on NO₂. Having said this the question remains if for publication in Nature Communications a scientific result is needed when, and there is no doubt, an innovative new technique is introduced. My answer is: yes, I would recommend publication, but this certainly is not a crystal clear case as it would be in a specialized Journal.

Reviewer #2 (Remarks to the Author):

The authors have addressed my comments properly. The associated changes to the manuscript indeed clarify the data acquisition scheme and help the data interpretation. Again, this manuscript presents a significant achievement in coherent spectroscopy by combining 2D electronic spectroscopy with mass detection. Despite the fact that NO₂ is a well-studied system, this new technique can still provide some new insights on the electronic processes and demonstrates the possibility of new directions of research. I therefore recommend this manuscript to be published in Nature Communications. There are just a few language issues that the authors may consider changing.

1. For the title of the manuscript, "2D" should be changed to "two-dimensional".
2. On page 5 of the current manuscript, the authors state: "Figure 2 demonstrates the simultaneous acquisition of reactant and product 2D spectra....". It is my understanding that the NO₂ is the reactant, while NO₂⁺ (parent ion) and NO⁺ (fragment ion) are the products. The authors may consider editing this sentence to avoid confusions.
3. The authors have added a new paragraph on Page 4 of the main text to emphasize that their new technique can offer new insights even in the multi-photon regime. This paragraph indeed helps the reading. The authors may consider emphasizing this point more in the abstract.

Reply to Reviewer Comments on Nature Communications

New Title: "Coherent two-dimensional electronic mass spectrometry"

Old Title: "Coherent 2D electronic mass spectrometry"

We again thank the reviewers for their positive evaluation of our work and address the remaining points below. We have modified the manuscript accordingly.

Reviewers' comments in this reply letter are printed in black, our response in blue, and explicit changes to the manuscript in red font.

Reviewer #1

Besides the general reply we were explicitly asked to comment on the criticisms of Reviewer#3 and the corresponding reply. The main point was that "a clear demonstration of some new advance in understanding" the studied system was missing. My own assessment in the first round went also in that direction: the scientific achievement (only) is the coupling to continuum states and the comparison with the Fano model; these, however, have not been presented convincingly. The long and extensive reply of the authors has clarified the situation on, I would say, all the raised issues by the reviewers and furthermore demonstrated that the presented work is on the highest scientific level and the authors have thoroughly analyze their data. However, the new scientific insight remains sober. Connected to the Fano model the authors clearly say that the work only demonstrates what "potentially can be learned from it" and, including all sections, I still cannot distil a clear answer on what has been learned on NO_2. Having said this the question remains if for publication in Nature Communications a scientific result is needed when, and there is no doubt, an innovative new technique is introduced. My answer is: yes, I would recommend publication, but this certainly is not a crystal clear case as it would be in a specialized Journal.

We thank the reviewer for this positive evaluation of our revisions.

Reviewer #2

The authors have addressed my comments properly. The associated changes to the manuscript indeed clarify the data acquisition scheme and help the data interpretation. Again, this manuscript presents a significant achievement in coherent spectroscopy by combining 2D electronic spectroscopy with mass detection. Despite the fact that NO₂ is a well-studied system, this new technique can still provide some new insights on the electronic processes and demonstrates the possibility of new directions of research. I therefore recommend this manuscript to be published in Nature Communications. There are just a few language issues that the authors may consider changing.

1. For the title of the manuscript, “2D” should be changed to “two-dimensional”.

We changed the manuscript title accordingly.

2. On page 5 of the current manuscript, the authors state: “Figure 2 demonstrates the simultaneous acquisition of reactant and product 2D spectra...”. It is my understanding that the NO₂ is the reactant, while NO₂⁺ (parent ion) and NO⁺ (fragment ion) are the products. The authors may consider editing this sentence to avoid confusions.

We agree with the referee that this sentence needs further specification and changed this section to “Figure 2 demonstrates the simultaneous acquisition of precursor and dissociation-product 2D spectra...”

3. The authors have added a new paragraph on Page 4 of the main text to emphasize that their new technique can offer new insights even in the multi-photon regime. This paragraph indeed helps the reading. The authors may consider emphasizing this point more in the abstract.

This is indeed worth being addressed in the manuscript abstract. However, we could not specify our multiphoton ionization experiments in more detail due to editorial guidelines regarding the length of the abstract and the aim to represent all of our findings equally. Nevertheless, we think that the phrase “Furthermore, we present 2D spectra of multiphoton ionization, disclosing distinct differences in the nonlinear response functions leading to the ionization products.” adequately describes our findings.